# Diurnal and seasonal molecular rhythms in human neocortex and their relation to Alzheimer's disease

Andrew S.P. Lim[1], Hans-Ulrich Klein[2,3], Lei Yu[4], Lori B. Chibnik[2,3], Sanam Ali[1], Jishu Xu[2,3], David A. Bennett[4] & Philip L. De Jager[2,3]

Circadian and seasonal rhythms are seen in many species, modulate several aspects of human physiology, including brain functions such as mood and cognition, and influence many neurological and psychiatric illnesses. However, there are few data regarding the genome-scale molecular correlates underlying these rhythms, especially in the human brain. Here, we report widespread, site-specific and interrelated diurnal and seasonal rhythms of gene expression in the human brain, and show their relationship with parallel rhythms of epigenetic modification including histone acetylation, and DNA methylation. We also identify transcription factor-binding sites that may drive these effects. Further, we demonstrate that Alzheimer's disease pathology disrupts these rhythms. These data suggest that interrelated diurnal and seasonal epigenetic and transcriptional rhythms may be an important feature of human brain biology, and perhaps human biology more broadly, and that changes in such rhythms may be consequences of, or contributors to, diseases such as Alzheimer's disease.

[1] Division of Neurology, Department of Medicine, Sunnybrook Health Sciences Centre, University of Toronto, 2075 Bayview Avenue, Room M1-600, Toronto M4N1X2, Ontario, Canada. [2] Program in Translational Neuropsychiatric Genomics, Department of Neurology, Brigham and Women's Hospital, Harvard Medical School, 77 Avenue Louis Pasteur, NRB 168c, Boston, Massachusetts 02115, USA. [3] Program in Medical and Population Genetics, Broad Institute, 415 Main Street, Cambridge, Massachusetts 02142, USA. [4] Rush Alzheimer's Disease Center and Department of Neurological Sciences, Rush University Medical Center, 600 South Paulina Street, Chicago, Illinois 60612, USA. Correspondence and requests for materials should be addressed to A.S.P.L. (email: andrew.lim@utoronto.ca).

Circadian and circannual rhythms are seen in many plant and animal species. Circadian rhythms modulate phenomena as diverse as bioluminescence in dinoflagellates[1] and cognitive function in humans[2,3], while circannual rhythms are seen in functions as diverse as flowering in plants[4] and hibernation in chipmunks[5], These rhythms also feature prominently in several human brain disorders. For example, prominent changes in circadian rest-activity[6] and body temperature rhythms[7] are seen in Alzheimer's disease, and there are seasonal rhythms of mood in seasonal affective disorder[8], of symptom onset in schizophrenia[9] and of human functional magnetic resonance imaging brain responses with cognitive tasks[10].

Notwithstanding the ubiquity of circadian and circannual rhythms—and their impact on human disease—there remain gaps in our understanding of the genetic and epigenetic mechanisms generating them and linking them to normal and abnormal tissue function, especially in the human brain. Mechanisms underlying circadian rhythms are better understood. An evolutionarily conserved transcriptional negative feedback loop lies at the core of the molecular clock in model organisms[11–13]; this mechanism influences tissue physiology by regulating the expression of tissue-specific sets of genes[14]. Underlying this circadian control of transcription are rhythms of transcription factor binding and histone modification[15–17]. In the human brain, diurnal rhythms of a large portion of the transcriptome have been revealed, and age- and depression-related differences have been reported[18,19]. However, there is a paucity of data concerning the effects of other human brain disorders such as Alzheimer's disease on the diurnal transcriptome. Moreover, there are few data regarding the relationship of diurnal rhythms of gene expression with rhythmic epigenetic modification in the human brain.

Less is known about the genetic and epigenetic mechanisms underlying seasonal rhythms, and how these molecular events influence tissue function. Seasonal rhythms of selected genes in specific brain regions have been reported in hamsters[20,21], ground squirrels[22] and songbirds[23]. Seasonal rhythms of DNA methylation may influence these rhythms[21,24] and seasonal rhythms of histone modification appear to be important in plants[25]. Recently, widespread seasonal rhythms of gene expression in human peripheral blood mononuclear cells have been reported[26]. However, seasonal rhythms of gene expression have never been demonstrated in any solid human organ, including the human brain. Moreover, the epigenetic regulation of these rhythms in human tissues is unknown, as is the extent to which they are altered by diseases such as Alzheimer's disease.

Using post-mortem human brain tissue obtained from two longitudinal cohort studies of ageing, we recently characterized large-scale diurnal rhythms of DNA methylation and their relation to diurnal rhythms of gene expression in the human dorsolateral prefrontal cortex[27], a brain region with prominent circadian rhythms of gene expression[18], and one that shows seasonal variation in human functional magnetic resonance imaging brain responses with cognitive tasks[10]. Building on these results, we obtained genome-wide RNA-sequencing (RNA-seq) and histone 3 lysine 9 acetylation chromatin immuno-precipitation sequencing (H3K9Ac ChIP-seq) data from overlapping sets of post-mortem human dorsolateral prefrontal cortex samples and examined, on a genome-wide scale, diurnal and seasonal rhythms of RNA expression, H3K9Ac and DNA methylation. We also characterized their interrelationship and their association with Alzheimer's disease.

Using these data, we demonstrate interrelated diurnal and seasonal rhythms of gene expression in the dorsolateral prefrontal cortex that are linked to parallel rhythms of epigenetic modification, associated with specific transcription factor-binding sites and altered in the context of Alzheimer's disease pathology. These data suggest that seasonal and diurnal molecular rhythms may play an important role in the biology of the human dorsolateral prefrontal cortex, and their disruption may be a potential contributor to, or consequence of, Alzheimer's disease.

## Results

**Diurnal/seasonal rhythms in the transcriptome and epigenome.** We studied post-mortem dorsolateral prefrontal cortex samples from 757 participants in two ongoing cohort studies of older persons, the Rush Memory and Ageing Project (MAP) and the Religious Orders Study (ROS), in which participants were free of dementia at study enrolment and agreed to annual evaluations and brain donation on death. Clinical characteristics of the study participants are in Table 1 and Fig. 1. Deaths were spread throughout the year and around the 24-h day (Fig. 2a) and we noted no relation between the date and time of death (Fig. 2b). We used RNA-seq to quantify dorsolateral prefrontal cortex expression of 18,709 autosomal GENCODE v14 genes and 42,873 autosomal GENCODE v14 isoforms expressed in at least 90% of our samples[27]. In parallel, we used the Illumina Infinium HumanMethylation450k Bead Chip Assay (Illumina, San Diego, CA) to assess DNA methylation at 420,132 autosomal cytosine-phosphate-guanine sites (CpGs)[28], and ChIP followed by DNA sequencing to assess H3K9Ac at 25,740 non-overlapping genomic regions spanning the autosomal genome[29]. The latter provide a truly epigenome-wide perspective that focuses on the parts of the genome that are actively transcribed.

To identify seasonal patterns in the expression of each gene, we considered expression levels as a function date of death relative to January 1. For diurnal patterns of gene expression, in keeping with similar studies[18,19], we considered expression levels as a

---

**Table 1 | Characteristics of the study participants*.**

| | Median (IQR) or number (%) | | |
| --- | --- | --- | --- |
| | **Participants with RNA-seq data ($n = 531$)** | **Participants with H3K9Ac ChIP-seq data ($n = 664$)** | **Participants with DNA methylation data ($n = 732$)** |
| Age (years) | 88.7 (84.5, 92.8) | 88.8 (84.3, 92.6) | 88.4 (83.9, 92.5) |
| Female sex | 334 (63%) | 433 (65%) | 466 (64%) |
| ≥1 Depressive symptoms | 325 (61%) | 408 (61%) | 455 (62%) |
| Post-mortem interval (h) | 5.7 (4.2, 8.2) | 5.8 (4.4, 8.3) | 5.8 (4.3, 8.5) |
| AD pathology summary score | 0.54 (0.15, 1.02) | 0.60 (0.16, 1.08) | 0.58 (0.15, 1.07) |
| NIA-Reagan Pathological Diagnosis of Alzheimer's disease | 315 (59%) | 408 (61%) | 441 (60%) |

*Please also see Fig. 1 for characteristics of the study participants. AD, Alzheimer's disease; ChIP, chromatin immunoprecipitation; H3K9Ac, histone 3 lysine 9 acetylation; IQR, interquartile range; NIA, National Institutes of Ageing; RNA-seq, RNA-sequencing.

function of time of death relative to sunrise on the date of death ('zeitgeber time', ZT). In secondary analyses, we repeated all analyses while considering time of death relative to (1) local clock time, which may be more reflective of the timing of artificial light exposure in industrialized societies and which shifts with daylight savings time, and (2) the midpoint of the dark period, which is relatively invariant across the seasons[30]. All three versions of our analyses returned similar results.

We found many genes, including several canonical circadian clock genes, that have robust diurnal and seasonal rhythms (Fig. 3 and Supplementary Figs 1 and 2). Based on visual inspection of these data and in keeping with prior work examining circadian[18,19] and seasonal[26] rhythms of human gene expression, we modelled these data as a sum of cosine curves with diurnal (period = 24 h) and seasonal (period = 1 year) periods.

For each of the 18,709 genes, we extracted the amplitude and acrophase of diurnal rhythmicity based on these cosine curves, and quantified the strength of diurnal rhythmicity by comparing models with and without terms for diurnal rhythmicity, and computing an $F$-statistic and $P$ value (Supplementary Data 1). We repeated these analyses for seasonal rhythmicity (Supplementary Data 1), and for diurnal and seasonal rhythms in mRNA isoform levels (Supplementary Data 2) as well as in our two levels of epigenomic data, the H3K9 acetylome (Supplementary Data 3) and in the DNA methylome (Supplementary Data 4).

Similar to other human brain data sets, where 8–12% of the transcriptome is diurnally rhythmic at $P < 0.05$ (refs 18,19), ~9% of the transcriptome (1,726 genes) in our data set was diurnally rhythmic at $P < 0.05$ ($F$-test, $n = 531$ samples). This set of genes was strongly enriched for genes previously reported as diurnally rhythmic in Brodmann's area 11 (ref. 19) ($\chi^2 = 18.0$, $P = 2.2 \times 10^{-5}$), Brodmann's area 47 (ref. 19) ($\chi^2 = 11.8$, $P = 5.9 \times 10^{-4}$) and dorsolateral prefrontal cortex[18] ($\chi^2 = 21.4$, $P = 3.7 \times 10^{-6}$) in other studies. The timing of clock gene expression in our data set was strongly correlated with the timing of clock gene expression in these other data sets (Fig. 3c and Supplementary Figs 1c and 2c; $R = 0.97$–0.99; $P < 0.0001$). These results illustrate the robustness of these rhythms across brain regions and data sets. They represent an important source of transcriptional variation in the brain, and our large data set behaves similarly to what has been observed previously, enabling us to connect our results with the existing framework of brain molecular rhythms.

We quantified the degree of diurnal rhythmicity across all 42,873 isoforms by computing the median $F$-statistic for diurnal rhythmicity. We compared this to the median $F$-statistic in 10,000 null data sets generated by randomly shuffling times of death to generate an empiric $P$ value. The transcriptome as a whole showed much greater diurnal rhythmicity than expected by chance (median $F = 1.02$, $P = 0.0252$, $n = 531$ samples; Fig. 4a). Similar analyses revealed significant seasonal rhythmicity in the transcriptome (median $F = 1.24$, $P = 0.0022$, $n = 531$ samples; Fig. 4b), diurnal rhythmicity in the H3K9 acetylome (median $F = 1.02$, $P = 0.0280$, $n = 664$ samples; Fig. 4c), and diurnal rhythmicity (median $F = 0.74$, $P < 0.0001$, $n = 732$ samples; Fig. 4e) and seasonal rhythmicity (median $F = 0.78$, $P < 0.0001$, $n = 732$ samples; Fig. 4f) in the DNA methylome. The degree of seasonal rhythmicity in the H3K9 acetylome also approached statistical significance (median $F = 0.95$, $P = 0.0526$, $n = 664$ samples; Fig. 4d).

When we limited these analyses to only those individual isoforms, H3K9Ac peaks and DNA methylation sites that were diurnally or seasonally rhythmic at $P < 0.05$ by the $F$-test (based on $n = 531$ samples for RNA, $n = 664$ samples for H3K9Ac, $n = 732$ samples for DNA methylation; Supplementary Data 2 and 3), or when we repeated these analyses in relation to local clock time or to the midpoint of the dark period, results were similar (Supplementary Figs 3–5).

**Relation of diurnal to seasonal rhythms.** By visual inspection, and in keeping with prior work[19,26], the distribution of diurnal and seasonal transcript acrophase times was bimodal. We used an empiric clustering approach based on self-organizing maps to classify transcripts into diurnal and seasonal clusters. This

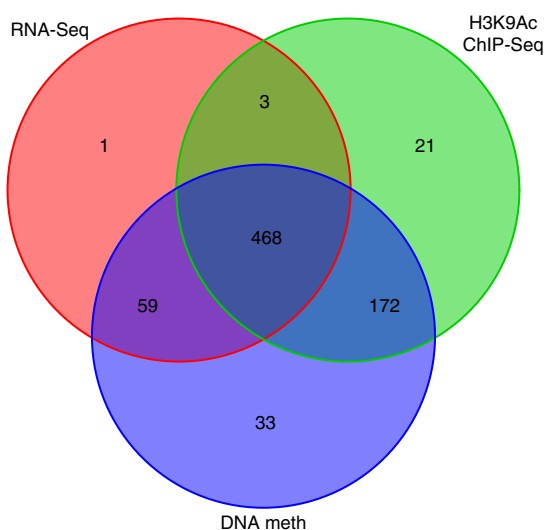

**Figure 1 | Subsets of samples with RNA-seq H3K9Ac and DNA methylation data.** Number of participants with RNA-seq, H3K9Ac ChIP-seq or DNA methylation data available, or combinations of the 3.

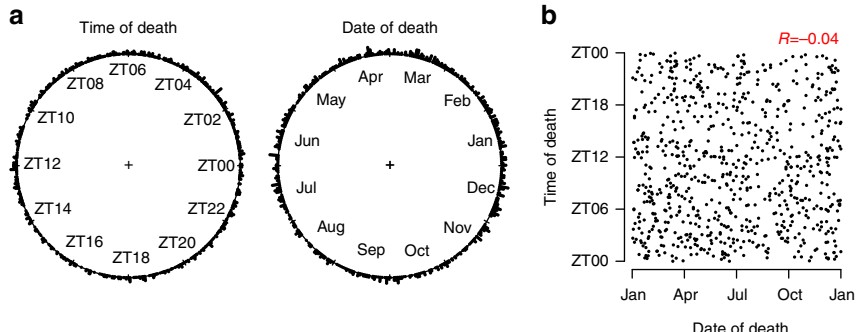

**Figure 2 | Distribution of times and dates of death. (a)** Distribution of times and dates of death for the samples used in this study. **(b)** Relationship between time and date of death. $n = 757$.

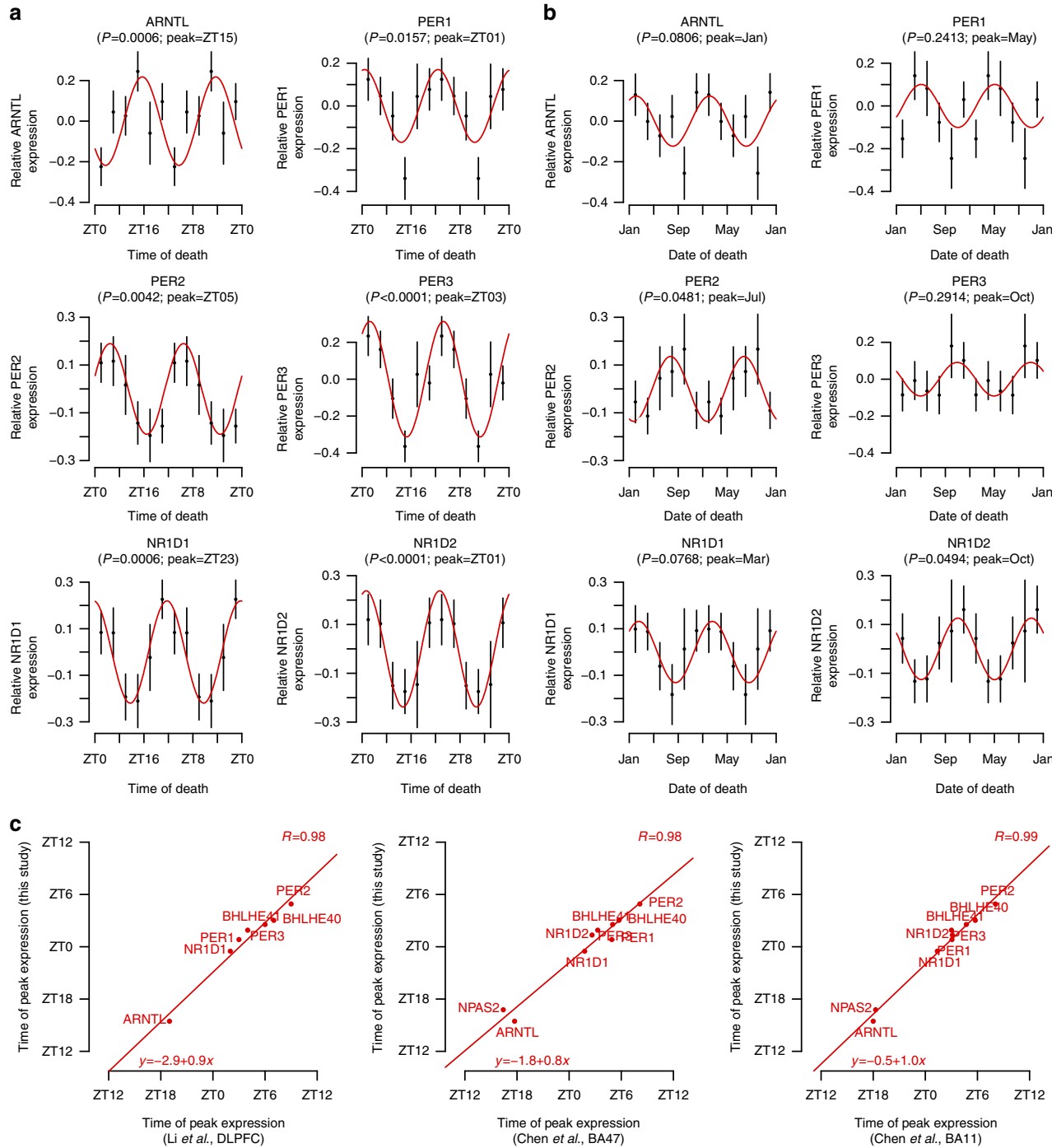

**Figure 3 | Diurnal and seasonal rhythms of clock gene expression.** (**a,b**) Relative expression by (**a**) time of death and (**b**) month of death for several genes known to be involved in the regulation of the mammalian circadian clock. Data plotted in (**a**) 4-h or (**b**) 2-month bins. ZT0 = sunrise. Dots indicate means and bars indicate s.e.'s of the mean. Data are double plotted. Red lines indicate fit cosine curve. *P* values for (**a**) diurnal or (**b**) seasonal rhythmicity are as calculated as described in the text using a model considering diurnal and seasonal rhythmicity concurrently, and adjusted for demographic and methodological covariates. (**c**) Correlation between the peak expression times of known circadian clock genes in our data set (*y*-axis) and published human prefrontal cortex data sets (*x*-axis). Red line indicates fit linear regression.

resulted in each transcript being classified into one of two empirically defined non-overlapping diurnal clusters, which were roughly centred about morning and evening, and one of two empirically defined non-overlapping seasonal clusters, which were roughly centred about spring and fall. This resulted in four distinct empirically defined sets of transcripts: evening/spring, evening/fall, morning/spring and morning/fall. By visual inspection, the timing of diurnal and seasonal transcript rhythms was closely linked, with morning-acrophase transcripts tending to have fall seasonal acrophases, and evening-acrophase transcripts tending to have spring seasonal acrophases (Fig. 4g). We confirmed this by computing a $\chi^2$-statistic and calculating an empiric *P* value in comparison with 10,000 permuted null data sets generated by shuffling dates and times of death. This association was strongly significant (Fig. 4h; $\chi^2 = 9,844.1$, $P < 0.0001$, $n = 531$ samples). A similar pattern was seen in the

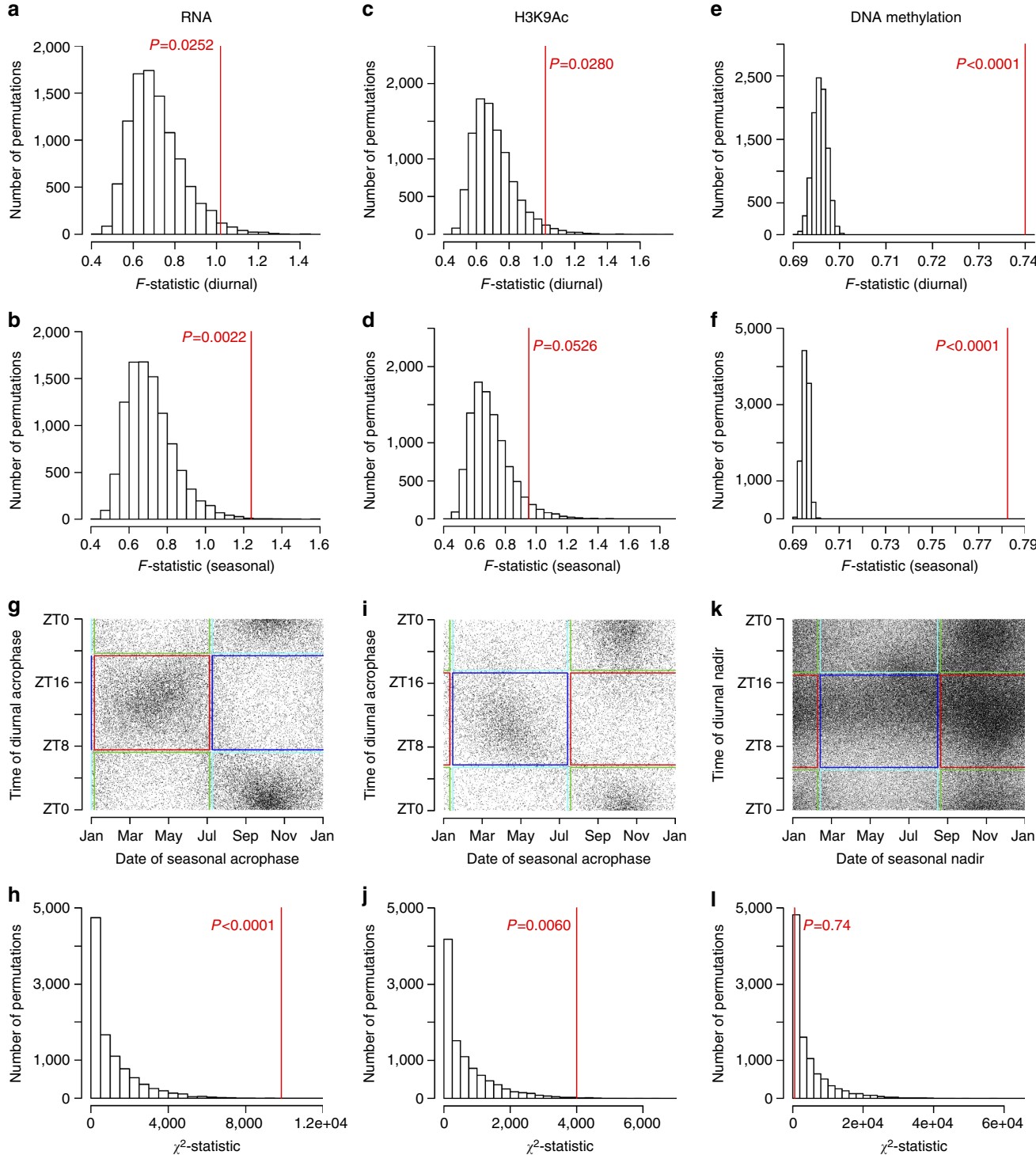

**Figure 4 | Diurnal and seasonal rhythmicity in the transcriptome and epigenome.** (**a**) Observed (red) versus expected (black) median *F*-statistic for diurnal rhythmicity considering all transcripts together. Null distribution estimated by consideration of 10,000 empiric null data sets generated by randomly shuffling times of death. (**b**) As in **a** but for seasonal rhythms. (**c,d**) As in **a,b** but for H3K9Ac peaks. (**e,f**) As in **a,b** but for DNA methylation sites. (**g**) Association between time of diurnal versus seasonal acrophases. Each dot represents a single transcript. Coloured boxes depict empirically derived clusters. (**h**) Observed (red line) versus expected (black bars) $\chi^2$-statistic for association between timing of diurnal and seasonal rhythms. Expected distribution empirically derived from 10,000 permuted null data sets generated by randomly shuffling times and dates of death. (**i,j**) Same as in **g,h** but for H3K9Ac peaks. (**k,l**) Same as in **g,h** but for the diurnal and seasonal nadirs of individual DNA methylation sites. See also Supplementary Figs 3–5.

H3K9 acetylome, with morning-acrophase H3K9Ac peaks tending to have fall seasonal acrophases, and evening-acrophase H3K9Ac peaks tending to have spring seasonal acrophases (Fig. 4i,j; $\chi^2 = 4,004.6$, $P = 0.0060$, $n = 664$ samples). For the DNA

methylome, we plotted the timing of the nadir rather than acrophase of methylation as hypomethylation rather than hypermethylation is classically associated with transcription. However, because the acrophase always follows the nadir by $\pi$

radians in a cosine curve, the analyses are statistically equivalent whether acrophase or nadir is used. The resulting patterns were different than those seen for the transcriptome and H3K9 acetylome. DNA methylation sites with fall nadirs were no more likely to have evening than morning nadirs (Fig. 4k,l; $\chi^2 = 882.5$, $P = 0.6662$, $n = 732$ samples). Similar results were seen when we limited these analyses to the most significant diurnally and seasonally rhythmic sites, or when we repeated these analyses in relation to local clock time or to the midpoint of the dark period (Supplementary Figs 3–5).

**Relation of the epigenomic rhythms to physical position.** We have previously shown that for rhythms of DNA methylation, diurnal acrophase time varies depending on distance from the nearest transcription start site (TSS)[27]. We investigated whether this was also true for diurnal rhythms of H3K9Ac by dividing H3K9Ac peaks into two groups—those within 2 kb of the TSS of transcripts expressed in more than 90% of our samples and those more than 2 kb away.

The timing of rhythms of H3K9Ac was associated with their proximity to active TSS (Fig. 5a–d). The angular distribution of diurnal acrophases for H3K9Ac peaks proximate to such sites differed from those for peaks distant from such sites ($W = 4,572.6$, $P = 0.0136$, $n = 664$ samples), and the difference in angular distribution of seasonal acrophases showed a trend toward significance ($W = 3,643.2$, $P = 0.0576$, $n = 664$ samples). Moreover, the set of H3K9Ac peaks proximate to active TSS was relatively enriched for morning/fall acrophase peaks, while the set of peaks distant from active TSS was relatively enriched for

evening/spring acrophase peaks ($\chi^2 = 4,841.0$, $P = 0.0125$, $n = 664$ samples). Similar results were seen when we limited these analyses to the most significant diurnally and seasonally rhythmic H3K9Ac peaks, or when we repeated these analyses in relation to local clock time or the midpoint of the dark period (Supplementary Figs 6,8a–d).

For DNA methylation, proximity to active TSS was associated with the timing of diurnal rhythms, as we had reported previously (Fig. 5e–h; $W = 3,8418.2$, $P = 0.026$, $n = 732$ samples). However, there was no association between proximity to an active TSS and the timing of seasonal rhythms ($W = 4,568.2$, $P = 0.7308$, $n = 732$ samples) or with the overall distribution of DNA methylation sites among the four temporal clusters ($\chi^2 = 36,324.6$, $P = 0.065$, $n = 732$ samples). Similar results were seen when we repeated these analyses in relation to local clock time or the midpoint of the dark period (Supplementary Figs 7 and 8e–h). However, when we limited these analyses to the most significant diurnally and seasonally rhythmic DNA methylation sites (Supplementary Fig. 6e–h), a relative enrichment of morning/fall nadirs was seen among sites proximate to active TSS ($\chi^2 = 354.5$, $P = 0.0001$, $n = 732$ samples).

**Relation between rhythms in the transcriptome and epigenome.** Both H3K9Ac and DNA methylation are thought to influence transcription. Therefore, we examined associations between the timing of rhythms of transcript abundance and the timing of rhythms of H3K9Ac and DNA methylation at nearby sites. To examine associations between diurnal transcript and H3K9Ac rhythms, we contrasted two groups of H3K9Ac peaks—those

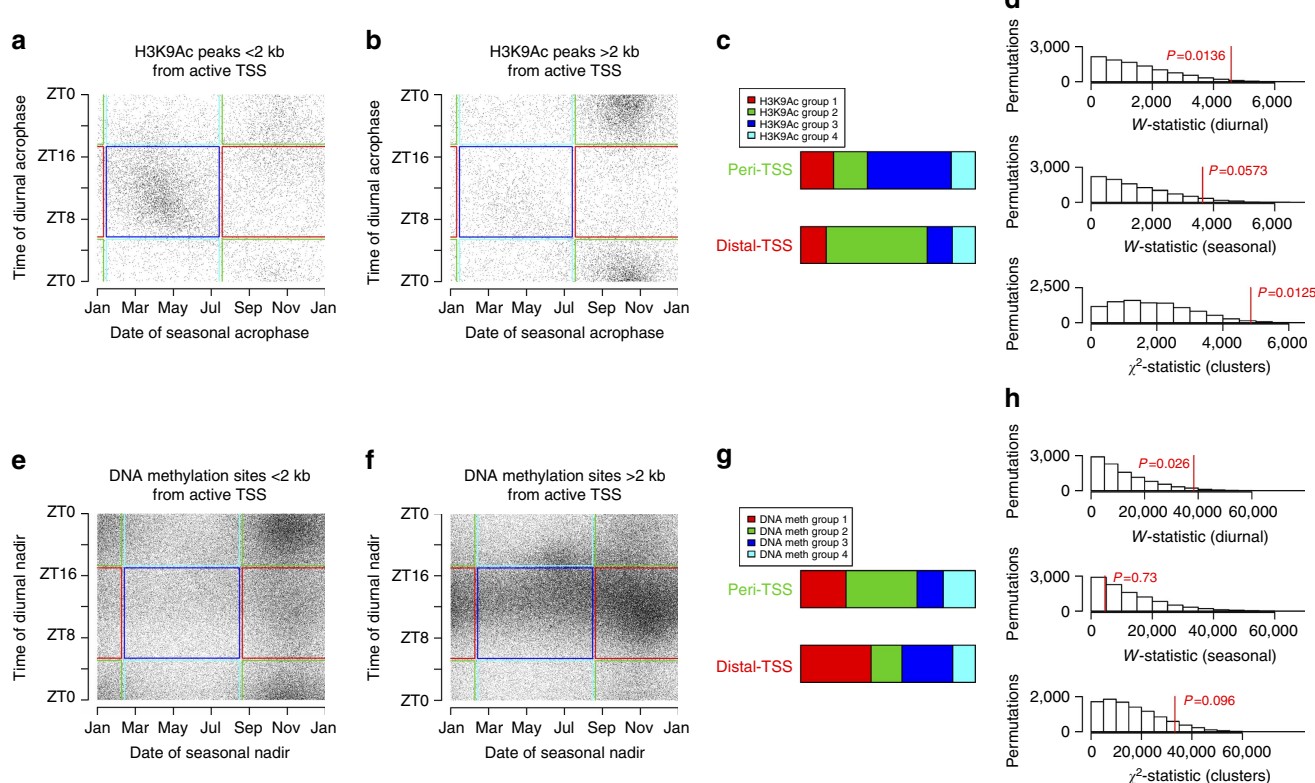

**Figure 5 | Physical position and diurnal and seasonal rhythms in the epigenome.** (**a**) Association between time of diurnal versus seasonal acrophase of H3K9Ac peaks <2 kb from active TSS. Each dot represents a single H3K9Ac peak. Coloured boxes depict empirically derived clusters. (**b**) Same as in (**a**), but for H3K9Ac peaks >2 kb from active TSS. (**c**) Temporal classification of H3K9Ac peaks less than or more than 2 kb from active TSS. (**d**) Observed (red line) versus expected distribution of W-statistic for angular distribution of diurnal acrophases, W-statistic for angular distribution of seasonal acrophases and $\chi^2$-statistic for temporal classification of H3K9Ac acrophases comparing sites < or >2 kb of active TSS. (**e–h**) Same as in (**a–d**) but for DNA methylation sites. See also Supplementary Figs 6–8.

within 2 kb of morning-acrophase transcripts and those within 2 kb of evening-acrophase transcripts. The angular distribution of diurnal acrophases for these two sets of H3K9Ac peaks differed (Fig. 6a,c; $W = 67.1$, $P < 0.0001$, $n = 471$ samples) with the set of H3K9Ac peaks near morning-acrophase transcripts being relatively enriched for morning-acrophase H3K9Ac peaks, and vice versa (Fig. 6e,g; $\chi^2 = 41.0$, $P = 0.0007$, $n = 471$ samples). Similarly, the angular distribution of seasonal acrophases for H3K9Ac peaks near spring versus fall-acrophase transcripts also differed (Fig. 6b,d; $W = 36.9$, $P = 0.0004$, $n = 471$ samples), and there was a trend toward relative enrichment for spring-acrophase H3K9Ac peaks near spring-acrophase transcripts, and fall-acrophase H3K9Ac peaks near fall-peaking transcripts (Fig. 6f,h; $\chi^2 = 16.8$, $P = 0.0770$, $n = 471$ samples).

We found similar results in the DNA methylome. The angular distribution of diurnal nadirs differed significantly between DNA methylation sites near the TSS of morning-acrophase transcripts and those near the TSS of evening-acrophase transcripts ($W = 857.6$, $P < 0.0001$, $n = 527$ samples; Fig. 6i,k) with a relative enrichment of morning-nadir DNA methylation sites near morning-acrophase transcripts, and a relative enrichment of evening-nadir DNA methylation sites near evening-acrophase transcripts (Fig. 6m,o; $\chi^2 = 665.3$, $P < 0.0001$, $n = 527$ samples). The angular distribution of seasonal nadirs also differed significantly between DNA methylation sites near the TSS of spring-acrophase transcripts and those proximate to the TSS of fall-acrophase transcripts (Fig. 6j,l; $W = 151.4$, $P < 0.0001$, $n = 527$ samples). However, here there was a relative enrichment of fall-acrophase (rather than fall-nadir) DNA methylation sites near fall-acrophase transcripts and spring-acrophase (rather than spring-nadir) DNA methylation sites near spring-acrophase transcripts (Fig. 6n,p; $\chi^2 = 74.7$, $P < 0.0001$, $n = 527$ samples).

Similar results were seen when we repeated these analyses in relation to local clock time or the midpoint of the dark period (Supplementary Figs 10 and 11). Qualitatively similar results were also seen when we limited these analyses to the most significantly rhythmic ($P < 0.05$ by the $F$-test based on $n = 527$ samples for DNA methylation and $n = 471$ samples for H3K9Ac) epigenetic sites proximate to the most significantly rhythmic ($P < 0.05$ by the $F$-test, $n = 527$ samples for DNA methylation and $n = 471$ samples for H3K9Ac) transcripts, although statistical significance was attenuated perhaps reflecting the smaller number of sites considered (Supplementary Fig. 9).

**Relation of rhythms to transcription factor-binding sites**. The local transcription factor environment has an important influence on the circadian timing of transcription in model systems[31], many canonical clock genes are themselves transcription factors, and transcription factor modification and processing (for example, phosphorylation, translocation and degradation) are important mechanisms regulating circadian rhythmicity. Therefore, we examined the impact of local transcription factor-binding sites on the timing of diurnal and seasonal rhythms of transcript expression, H3K9Ac and DNA methylation using genome-wide-annotated binding sites for 161 transcription factors from the ENCODE project[32–34]. We considered a TSS, H3K9Ac peak or DNA methylation site to be locally associated with a transcription factor if it overlapped with one of its ENCODE-annotated binding sites, or was within 2 kb of it. We used logistic regression models to examine the independent impact of the local presence of binding sites for each of the 161 ENCODE transcription factors on the odds of having a spring versus fall or evening versus morning transcript, H3K9Ac peak or DNA methylation site.

Of the 161 transcription factors examined, 11 had binding sites that were associated with the timing of seasonal or diurnal

acrophases in the transcriptome, H3K9 acetylome or DNA methylome at a false discovery rate (FDR) $< 0.05$ (Fig. 7a). Effect estimates were similar irrespective of whether they were estimated on the basis of all transcripts/H3K9Ac peaks/DNA methylation sites, or only those that displayed the strongest rhythmicity (Supplementary Fig. 12). Their estimated effects on diurnal and seasonal rhythms were tightly correlated ($R = 0.95$, $P < 0.0001$ for RNA rhythms; $R = 0.77$, $P = 0.0058$ for H3K9Ac rhythms; $R = 0.8$, $P = 0.0032$ for DNA methylation rhythms; Fig. 7b–d). Their estimated effects on diurnal RNA and DNA methylation rhythms were strongly correlated ($R = 0.83$, $P = 0.0015$; Fig. 7f), and their effects on diurnal RNA and H3K9Ac rhythms were weakly anticorrelated ($R = -0.57$, $P = 0.065$; Fig. 7e). In contrast, their effects on the timing of seasonal epigenetic and transcriptomic rhythms were not clearly linked (Fig. 7g,h). Similar results were seen when we repeated these analyses in relation to local clock time or the midpoint of the dark period (Supplementary Figs 13 and 14).

**Relation to Alzheimer's disease**. Alzheimer's disease[35] is associated with differences in the timing of physiological markers of circadian rhythmicity. To examine the association between Alzheimer's disease pathology and diurnal and seasonal rhythms of RNA expression, H3K9Ac and DNA methylation, we augmented our models with terms to capture differences in phase and amplitude between those with and without pathologically defined Alzheimer's disease, reflecting the state of the brain tissue itself, which is the source of the transcriptomic and epigenomic data, while adjusting for the effects of age and sex on these parameters. Compared to individuals without pathologic Alzheimer's disease, the diurnal transcript rhythms in individuals with pathologic Alzheimer's disease were advanced by nearly 1.5 h, while their seasonal rhythms were delayed by approximately half a month (Fig. 8a–c). These differences were particularly pronounced for morning/fall- and evening/spring-acrophase transcripts and were less prominent for morning/spring- and evening/fall-acrophase transcripts, indicating a differential effect on different temporal classes of transcripts (Supplementary Fig. 15). Differences in rhythmic H3K9Ac were less pronounced. No significant differences in phase and amplitude between those with and without Alzheimer's disease were found when we considered all H3K9Ac peaks, although there were trends toward a slight delay and attenuation of diurnal rhythms (Fig. 8d–f). When individual temporal classes of H3K9Ac peaks were examined, diurnal rhythms of evening/fall-acrophase H3K9Ac peaks were both attenuated and delayed by nearly 1.5 h, with minimal changes in the other clusters of H3K9Ac peaks (Supplementary Fig. 15). Differences in DNA methylation rhythms were also modest, with a slight attenuation of the amplitude of diurnal rhythms (Fig. 8g–i), in keeping with our prior report[27], particularly for evening-nadir DNA methylation sites (Supplementary Fig. 15). Phase differences were modest when sunrise was used as the reference time (Fig. 8g–i). However, when we used clock time (as in our prior work) or the midpoint of the dark period as the reference times (Supplementary Figs 18g–i and 19g–i), there was a trend towards a delay in brains with pathologically defined Alzheimer's disease, in keeping with our prior report[27]. These findings were qualitatively similar when we considered only these transcripts, H3K9Ac peaks and DNA methylation sites with $P < 0.05$ for both diurnal and seasonal rhythmicity (Supplementary Figs 16 and 17)

## Discussion

Genome-scale analysis of over 750 post-mortem human dorsolateral prefrontal cortex samples revealed the presence of diurnal

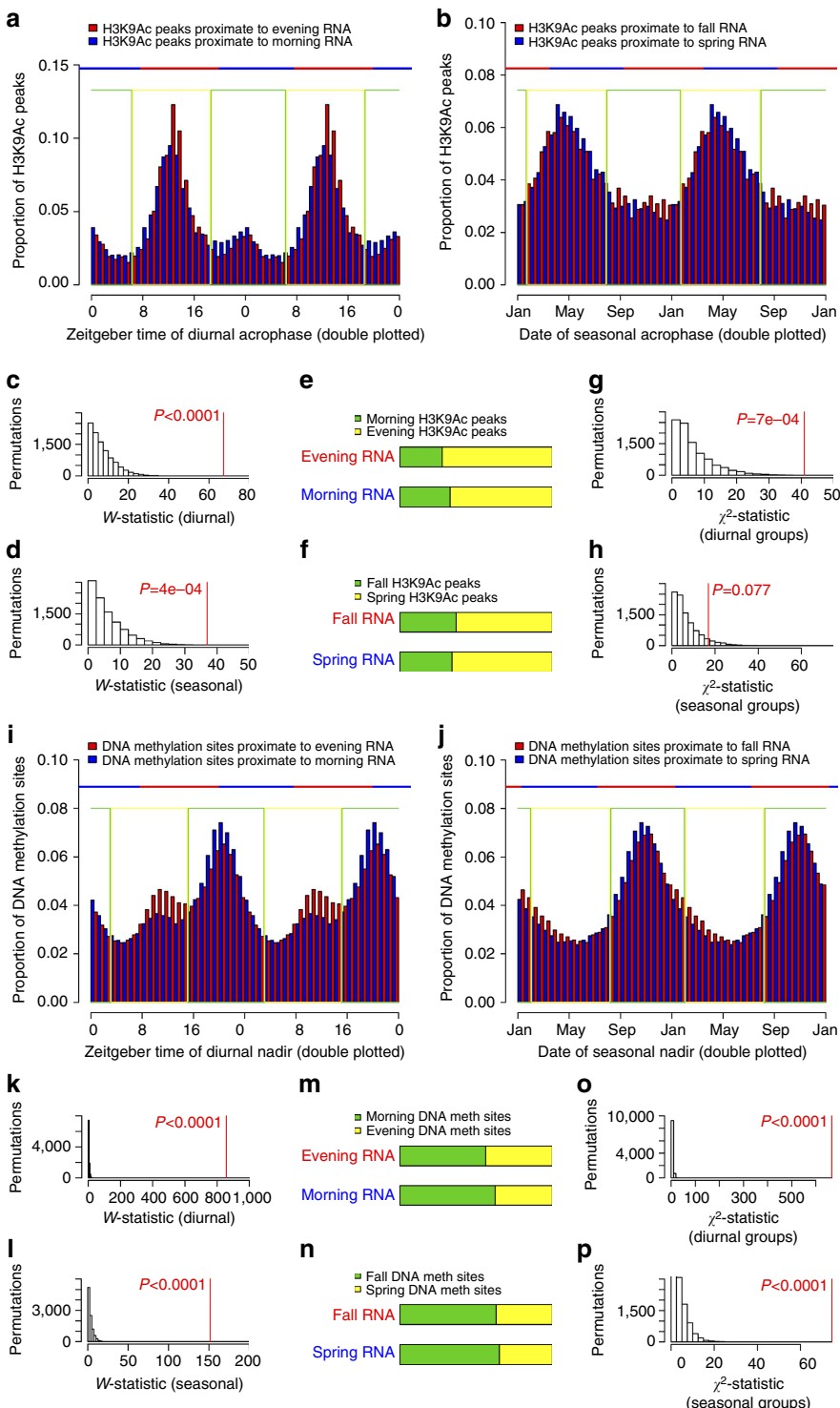

**Figure 6 | Association between rhythms in the transcriptome and epigenome. (a)** Temporal distribution of H3K9Ac diurnal acrophases for peaks within 2 kb of the TSS of evening acrophase (red) and morning acrophase (blue) transcripts. Data are double plotted. Horizontal line indicates temporal boundaries of the associated transcript classes. Green and yellow boxes indicate temporal boundaries of H3K9Ac classes (evening yellow, morning green). **(b)** Same as in (**a**) but for seasonal acrophases. **(c)** Observed versus expected *W*-statistic comparing the diurnal distributions of H3K9Ac acrophases for peaks within 2 kb of the TSS of evening-peaking and morning-peaking transcripts. Expected distribution is derived from 10,000 permuted null data sets generated by randomly shuffling times of death. **(d)** Same as in (**c**) but for seasonal H3K9Ac acrophases. **(e)** Diurnal temporal classification of H3K9Ac peaks near evening- versus morning-peaking transcripts. **(f)** Same as in (**e**) for seasonal temporal classification. **(g)** Observed versus expected $\chi^2$-statistic comparing the diurnal temporal classification of H3K9Ac peaks near evening- versus morning-peaking transcripts. **(h)** Same as in (**g**) but for seasonal temporal classification. **(i–p)** Same as in (**a–h**) but for the diurnal and seasonal nadirs of DNA methylation sites near evening- versus morning-, or fall- versus spring-peaking transcripts. See also Supplementary Figs 9–11.

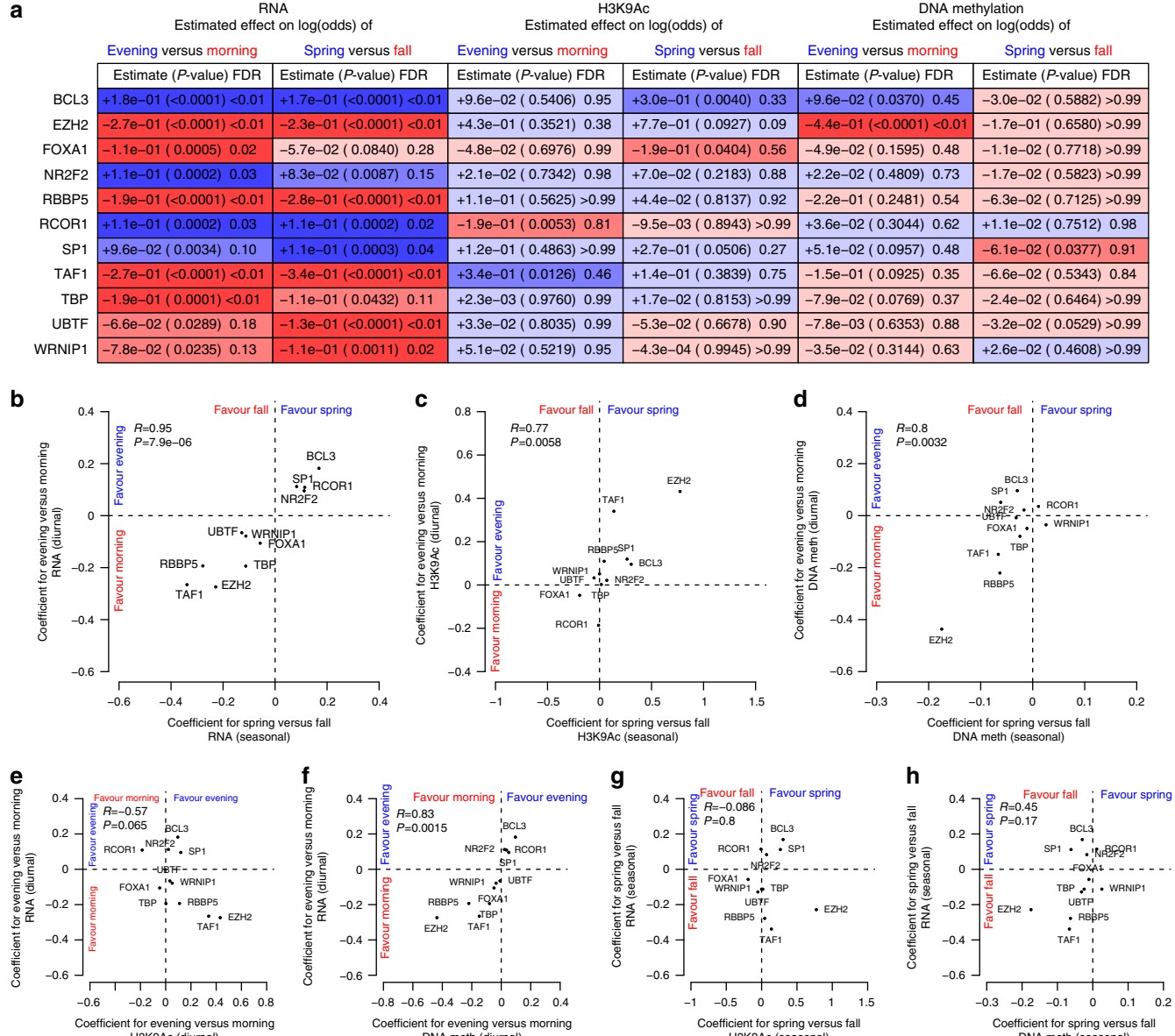

**Figure 7 | Transcription factor-binding sites and rhythms in the transcriptome and epigenome.** (**a**) Transcription factor-binding sites associated with at least one of diurnal or seasonal rhythms of RNA expression, H3K9 acetylation or DNA methylation at FDR < 0.05. Red indicates favours morning or fall timing; blue indicates favours evening or spring timing. Dark shading indicates significant at analysis FDR < 0.05. Medium shading indicates significant at nominal $P < 0.05$. Light shading indicates nominal $P > 0.05$. (**b**) Estimated coefficients for log(odds) of morning-acrophase RNA versus coefficients for log(odds) of fall-acrophase RNA. (**c,d**) Same as in (**b**) but for acrophases of H3K9Ac peaks (**c**) or nadirs DNA methylation sites (**d**). (**e**) Estimated coefficients for log(odds) of morning-acrophase RNA versus estimated coefficients for log(odds) of morning-acrophase H3K9Ac peaks. (**f**) Same as in (**e**), but for seasonal rhythms. (**g**) Estimated coefficients for log(odds) of morning-acrophase RNA versus estimated coefficients for log(odds) of morning-nadir DNA methylation sites. (**h**) Same as in (**g**) but for seasonal rhythms. See also Supplementary Figs 12–14.

and seasonal rhythms in the human brain transcriptome, H3K9 acetylome and DNA methylome. The timing of diurnal and seasonal rhythms were related, suggesting shared regulatory mechanisms, and were strongly influenced by the local transcription factor-binding site environment, with shared groups of transcription factors associated with both diurnal and seasonal rhythms. Similarly, rhythms in the transcriptome and epigenome were temporally linked and associated with shared transcription factor-binding sites. Finally, differences in rhythms in the transcriptome and epigenome accompanied a pathological diagnosis of Alzheimer's disease, suggesting that they may be an important contributor to, or consequence of, Alzheimer's disease pathology.

Using a time of death analytic approach similar to the present study, two recent studies have reported diurnal rhythms of the expression of several hundred genes in several human brain regions[18,19]. Our results are largely concordant with these, both with regard to the overall bimodal clustering of diurnal acrophases and with regard to the specific timing of canonical circadian clock genes. The patterns of clock gene expression in our study are also concordant with our previous study in which we used microarrays to assess clock gene expression[36]. Moreover, the set of genes identified as diurnally rhythmic in our data was strongly enriched for genes identified as diurnally rhythmic in two recent studies[18,19]. We extend these results by showing concurrent seasonal rhythms of gene expression, which had to

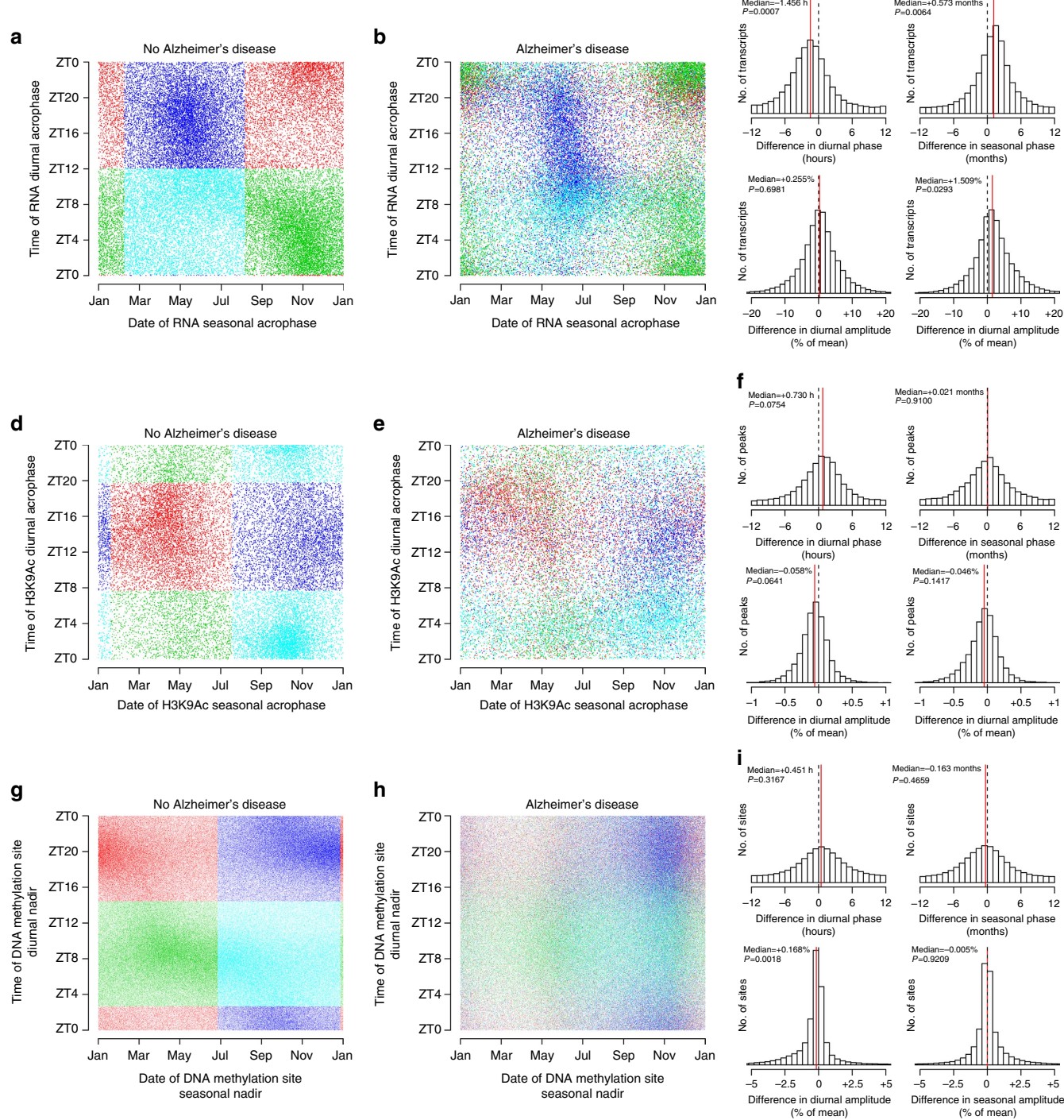

**Figure 8 | Alzheimer's disease and diurnal/seasonal rhythms in the transcriptome and epigenome.** (**a**) Model-predicted times of diurnal versus seasonal acrophases in samples without a pathological diagnosis of Alzheimer's disease. Each dot represents a single transcript. Colours depict empirically derived clusters. (**b**) As in (**a**) but for samples with a pathological diagnosis of Alzheimer's disease. Colours depict clustering based on samples without Alzheimer's disease. (**c**) Distribution of differences in timing and amplitude of transcript rhythms for samples with Alzheimer's disease versus samples without. Red lines indicate median differences. *P* values calculated by comparison to 10,000 empiric null data sets generated by randomly shuffling times and dates of death. (**d**–**f**) As in (**a**–**c**) but for H3K9Ac peaks. (**g**–**i**) As in (**a**–**c**) but for DNA methylation sites. See also Supplementary Figs 15–19.

our knowledge never before been demonstrated in any human solid organ. Like a recent study of seasonal rhythms of gene expression in human peripheral blood mononuclear cells and adipocytes[26], we found a bimodal pattern of seasonal gene expression. Interestingly, the peaks did not correspond to the peak and trough of seasonal photoperiod. Rather, they were

similar to a prior study in which the clusters of seasonal gene acrophases lagged the solstices by 1–2 months[26]. This argues against photoperiod being the primary driver of these rhythms and suggests that other environmental (for example, temperature, whose peaks and troughs lag photoperiod in temperate climates) or behavioural factors (for example, physical activity), or even an

endogenous circannual clock, as has been demonstrated in the European hamster[20], may be influencing these rhythms. Alternatively, these acrophases of gene expression may reflect the rate of change of photoperiod, which is fastest in the spring and summer. A recent study demonstrated that the human dorsolateral prefrontal cortex perfusion response to certain cognitive tasks displays a seasonal acrophase in the fall, tracking the rate of change of photoperiod[10].

We previously reported diurnal rhythms of DNA methylation in the human dorsolateral prefrontal cortex. To our knowledge, circadian rhythms of histone modification previously have not been examined in the human brain. In this study, we found prominent diurnal rhythms of H3K9Ac, similar to those previously reported in mice[15]. In addition, we found seasonal rhythms of H3K9Ac and DNA methylation, which have not, to our knowledge, been previously demonstrated in any mammalian tissue. Moreover, the timing of diurnal and seasonal rhythms varied with proximity to active TSS. This suggests that the timing of diurnal and seasonal regulation of H3K9Ac and DNA methylation may depend in part on the local transcriptional state and is in keeping with our previous finding that the timing of diurnal rhythms of DNA methylation depends on proximity to nearby active TSS[27].

In model organisms, diurnal rhythms of histone modification and diurnal rhythms of transcription are related[15–17]. We found similar relationships in the human dorsolateral prefrontal cortex with regions near expressed transcripts being relatively enriched for H3K9Ac peaks with acrophases and DNA methylation sites with nadirs coincident with or preceding peak transcript abundance. When we examined seasonal rhythms, a similar enrichment was seen, although here the enrichment was for DNA methylation acrophases coincident with, and H3K9Ac acrophases several months preceding, peak transcript abundance. This delay may suggest that although seasonal rhythms of H3K9Ac and DNA methylation may contribute to seasonal rhythms of transcript expression, other more proximate causal factors must also be contributing (for example, other epigenetic changes, derepression of repressors or other factors).

We found a striking relationship between the timing of diurnal and seasonal rhythms for gene expression and H3K9Ac as well as, to a lesser degree, DNA methylation. This suggests that shared mechanisms regulate diurnal and seasonal rhythmicity in the human dorsolateral prefrontal cortex. Indeed, overlap between mechanisms regulating circadian and circannual rhythmicity has been reported at the anatomic[37–39] and genetic levels[40]. In an exploratory analysis in this study, the effects of the local transcription factor-binding site environment on the timing of diurnal and seasonal rhythms overlapped, with the magnitude and direction of the effect of a transcription factor-binding site on diurnal rhythms predicting the magnitude and direction of its effect on seasonal rhythms. Furthermore, the effect of a transcription factor-binding site on diurnal DNA methylation rhythms was a better predictor of its effect on transcript rhythms than its effect on H3K9Ac rhythms, raising the possibility that DNA methylation rhythms may mediate the link between specific transcription factors and diurnal and seasonal transcript rhythms. It is noteworthy that of the 11 transcription factors significantly associated with the timing of diurnal or seasonal rhythms, four transcription factors (EZH2 (ref. 41, RBBP5 (refs 42,43), SP1 (ref. 44) and TBP[45]) have previously been implicated in the circadian machinery, reported to bind to members of the circadian machinery or found to be enriched in the promoter regions of clock-controlled genes. The role of the remaining seven genes in regulating seasonal or diurnal rhythms merits further study. One caveat in interpreting these results is that our binding site annotations were taken from human cell lines. We are not aware of genome-wide data sets using transcription factor ChIP-seq to generate similar annotations for human neocortex; the results of our hypothesis-generating analysis will guide targeted experiments to generate such data and will also guide in vitro validation efforts in human in vitro models, with a goal of further elucidating transcription factors regulating diurnal and seasonal epigenetic and transcriptional rhythms in the human brain.

Concordant with our previous report[27], a pathological diagnosis of Alzheimer's disease was associated with attenuated diurnal rhythms of DNA methylation. It was also associated with attenuated diurnal rhythms of H3K9Ac in some but not other classes of H3K9 peaks, while no attenuation of diurnal transcript rhythms was seen. The selective attenuation of rhythms on some but not other clusters of H3K9Ac peaks and DNA methylation sites, and the sparing of the amplitude of diurnal transcript rhythms argues against the possibility that these effects reflect greater within-group diurnal dyssynchrony in individuals with Alzheimer's disease pathology compared to those without, which has been suggested as an explanation for the attenuation of diurnal transcript rhythms associated with depression[18]. If this were the case, it should affect all diurnal rhythms equally.

Compared to a prior analysis of a subset of these data[27], we found a less robust effect of Alzheimer's disease pathology on the phase of diurnal methylation rhythms once we adjusted for seasonal effects. This reflects in part the different reference time frame (midnight clock time versus sunrise time) used in the two analyses. Indeed, in secondary analyses, when we used clock time as the reference time, the phase delay associated with Alzheimer's disease pathology approached statistical significance. In addition, given the strong association between seasonal and diurnal rhythms that we have shown here, it is also likely that in the previous study, there was some confounding of diurnal effects by seasonal effects, highlighting the importance of adjusting for seasonal effects when examining diurnal rhythms.

Although we found that pathologic Alzheimer's disease had modest effects on the timing of diurnal or seasonal epigenetic rhythms after controlling for other covariates, the diurnal transcript rhythms of individuals with Alzheimer's disease pathology were phase advanced by almost 1.5 h and their seasonal rhythms delayed by almost half a month. There are at least two possible explanations for this observation. First, rhythms of transcription reflect not only DNA methylation and H3K9Ac but also other epigenetic marks, and other transcriptional events such as RNA polymerase binding[17,46,47]. Alzheimer's disease pathology may alter the phase relationship between rhythms of H3K9Ac, DNA methylation and RNA transcription by affecting these other processes. Second, post-transcriptional processes appear to play an important role in determining the timing of rhythms of transcript abundance[17,46,47]. MicroRNAs may play an important role here[48] as may rhythms of alternative splicing[49]. Alzheimer's disease pathology may alter post-transcriptional RNA processing, further decoupling rhythms of H3K9Ac and DNA methylation from rhythms of transcript abundance.

In considering these data, a few methodological points are worth noting. As in other recent studies[18,19,26], we inferred group-level average diurnal and seasonal rhythms. Because ethical considerations preclude serial sampling of neocortical tissue from living human subjects, we only had one sample per individual, making exploration of individual level differences impossible. In addition, we had no data on behavioural state (awake versus asleep), medical status, environmental conditions (light, temperature) or activities proximate to death. Therefore, we cannot distinguish diurnal or seasonal rhythms driven by intrinsic circadian or circannual clocks, from those reflective of rhythms in environmental conditions, social environment,

behaviour or medical health. Third, humans experience both natural and artificial light, the timing of which may change by season. Moreover, sleep timing, which is a major influence on exposure to darkness, is linked to clock time and associated social schedules rather than natural light exposure. These considerations complicate the selection of an appropriate reference time for diurnal rhythms, and also the interpretation of diurnal and seasonal rhythmicity. However, our results were robust: results were consistent between our primary analytic approach using sunrise time (reflective of the timing of natural light exposure) as the reference time for diurnal analyses, and our secondary analyses using clock time (linked to artificial light, and to the timing of sleep and social schedules) or the midpoint of the dark period (invariant across seasons) as the reference time. Finally, although this is the largest study to date to examine diurnal rhythms of gene expression in the human brain and it provided ample statistical power for group-level comparisons, our study lacked statistical power to draw firm conclusions about diurnal and seasonal rhythms at the individual chromosomal locus level, and, therefore, we did not pursue locus-level analyses.

This study also had a number of methodological strengths. We measured transcript expression, H3K9Ac and DNA methylation genome-wide in the same samples, allowing the inference of the temporal relationships between them. In addition, data were obtained at all circadian times and dates of death, and at a relatively high temporal density, allowing greater precision than would be provided by less frequent sampling. Moreover, as all participants were organ donors, both time and date of death were relatively accurately determined, and post-mortem intervals were short. Also, we simultaneously considered both diurnal and seasonal effects in the same model, allowing us to characterize their independent contributions. Further, the participants were clinically well characterized, allowing us to examine the impact of key brain disorders like pathologic Alzheimer's disease, on human dorsolateral prefrontal cortex diurnal and seasonal genetic and epigenetic rhythms. Finally, the brain tissue was obtained from two prospective cohort studies with extraordinarily high rates of follow-up and autopsy, minimizing bias due to selective attrition.

Considered together, these results highlight a multilevel architecture of interrelated diurnal and seasonal rhythms of gene expression and epigenetic modification in the human dorsolateral prefrontal cortex, with shared regulatory mechanisms. Moreover, they suggest that changes in the amplitude, timing or phase relationships of these molecular rhythms may be contributors to, or consequences of, Alzheimer's disease pathology. Larger human studies sampling multiple brain regions, or even multiple organs, are needed to provide locus-level spatial resolution while linking these molecular diurnal and seasonal events to key nodes (such as the suprachiasmatic nucleus) thought to play roles in circadian and circannual rhythmicity. Meanwhile, work *in vitro* and in model organisms is needed to better delineate regulatory mechanisms suggested by our analyses.

## Methods

**Study participants.** Data from participants in two ongoing longitudinal cohort studies of older persons were included in this study: the Rush MAP and the ROS. The ROS is a longitudinal study of ageing in Catholic brothers, nuns and priests from across the United States[50]. The MAP is a community-based study of ageing in the greater Chicago area[51]. Participants in both studies are free of known dementia at study enrolment, and agree to annual evaluations and brain donation on death. At the time of the current analyses, 680 ROS participants and 726 MAP participants had died and had autopsies performed. Because all ROS and MAP participants are organ donors, date and time of death are generally well captured in the vast majority of participants. We excluded from further analysis data from 215 dorsolateral prefrontal cortex samples where time or date of death or other clinical covariate data (see below) were not available.

Of the 1191 samples for whom these data were available, dorsolateral prefrontal cortex RNA expression data passing quality control criteria (see below) were available from 531 samples, DNA methylation data passing quality control criteria (see below) were available from 732 samples and H3K9Ac ChIP-seq data passing quality control criteria (see below) were available from 664 samples. Data from these participants were included in the current analyses. Their characteristics are shown in Table 1 and the overlap between these sets is shown in Fig. 1.

**Statement of ethics approval.** This study was approved by the Institutional Review Board of Rush University Medical Centre and was conducted in accordance with the latest version of the Declaration of Helsinki. All participants provided written informed consent, and an Anatomic Gift Act for organ donation.

**Evaluation of transcript expression.** RNA was extracted from dorsolateral prefrontal cortex blocks with the miRNeasy Mini Kit (Qiagen, Venlo, The Netherlands) and the RNase-free DNase Set (Qiagen). RNA concentration was quantified by Nanodrop (Thermo Fisher Scientific, Waltham, MA, USA) and an Agilent Bioanalyser was used to assess quality. Samples from which <5 µg of RNA were obtained, or samples with a Bioanalyser RNA integrity score of 5 or less, were excluded from further analysis. The strand-specific dUTP method[52] with poly-A selection[53] was used by the Broad Institute Genomics Platform to prepare the RNA-seq library. Poly-A selection was followed by first-strand-specific cDNA synthesis, with dUTP used for second-strand-specific cDNA synthesis, followed by fragmentation and Illumina adapter ligation for library construction. An Illumina HiSeq machine was used to perform sequencing with 101 bp paired-end reads, achieving a coverage of 150M reads for the first 12 samples, which served as a deep coverage reference. The remaining samples were sequenced with coverage of 50M reads. Next, beginning and ending low-quality bases and adapter sequences were trimmed from the reads, and ribosomal RNA reads were removed. The Bowtie 1 software package[54] was used to align the trimmed reads to the reference genome. Finally, the RSEM software package was used to estimate, in units of fragments per kilobase per million mapped fragments (FPKMs), expression levels for 55,889 individual GENCODE v14 genes, and 190,051 distinct GENCODE v14 isoforms. These data are available through the synapse.org AMP-AD Knowledge Portal (http: //www.synapse.org; SynapseID syn3388564). We analysed only genes and transcripts on autosomal chromosomes, and excluded genes and transcripts expressed in <90% of our samples, leaving 18,709 individual GENCODE v14 genes and 42,873 individual GENCODE v14 isoforms in these analyses. All FPKM values were log-transformed before further analysis. Principal component analysis was used to assess sample quality, and only those samples with principal components 1, 2 and 3 values within 3 s.d. from their respective means were included. Data from 531 samples meeting quality control criteria as above, and with full clinical data as described below, were included in this analysis.

**Evaluation of DNA methylation.** We assessed DNA methylation in 746 dorsolateral prefrontal cortex samples as described previously[27,28]. Frozen 100 mg dorsolateral prefrontal cortex blocks were thawed on ice and grey matter manually dissected and the QIAamp DNA Mini Kit (Qiagen) was used to extract DNA. The Quant-iT PicoGreen Kit (Life Technologies, Carlsbad, CA, USA) was used to measure DNA concentration. The Illumina Infinium HumanMethylation450k Bead Chip Assay (Illumina) was used by the Broad Institute's Genomics Platform (Cambridge, MA, USA) to assay 16 µl of DNA from each sample at a concentration of 50 ng µl$^{-1}$. The Methylation Module v.1.8 from the Genome Studio software suite (Illumina) was used to carry out colour channel normalization and background removal, and to generate $\beta$ values and detection $P$ values for 485,513 CpG site across the human genome. These data are available through the synapse.org AMP-AD Knowledge Portal (http: //www.synapse.org SynapseID syn3157275). Probes with detection $P$ values >0.01 in any samples, with 47/50 nucleotides matching sex chromosome sequences during sequence alignment with BLAT, in which a single-nucleotide polymorphism with a minor allele frequency ≥0.01 exists within 10 bp upstream or downstream of the CpG site, or on a sex chromosome were excluded from further analysis. This left 420,132 autosomal CpGs in the data set. Principal component analysis was used to assess sample quality, and only those samples with principal components 1, 2 and 3 values within 3 s.d. from their respective means were included. As well, samples in which at least 2 out of the 10 bisulfite conversion control probes failed to reach a value of 0.8 were excluded. Data from 732 samples meeting quality control criteria as above, and with full clinical data as described below, were included in these analyses.

**Evaluation of H3K9Ac.** Grey matter was dissected on ice from blocks of dorsolateral prefrontal cortex, minced and crosslinked with 1% formaldehyde at room temperature for 15 min and quenched with 0.125 M glycine. After homogenizing the tissue in cell lysis buffer using the Tissue Lyser and a 5 mm stainless-steel bead, the nuclei were lysed in cell lysis buffer and chromatin was sheared by sonication. Chromatin was incubated overnight at 4 °C with the Millipore anti-H3K9Ac monoclonal antibodies (catalogue no. 06-942, lot no.: 31636) and purified with protein A sepharose beads. Finally, extracted DNA was used for Illumina library construction following usual methods and sequenced with

36 bp single-end reads on the Illumina HiSeq. Reads were aligned by the BWA algorithm against the human reference genome GRCh37 (ref. 55). MACS2 was applied for peak detection to each sample separately using the broad peak option and a $q$ value cutoff of 0.001 (ref. 56). Pooled genomic DNA of seven samples was used as negative control. These data are available through the synapse.org AMP-AD Knowledge Portal (http://www.synapse.org SynapseID syn4896408). Five different quality filters were used to remove low-quality samples[57]: (i) $\geq 15 \times 10^6$ unique reads, (ii) non-redundant fraction $\geq 0.3$, (iii) cross-correlation $\geq 0.03$, (iv) fraction of reads in peaks $\geq 0.05$ and (v) $\geq 6,000$ peaks. After quality control, 669 out of 712 samples remained. H3K9Ac domains were defined by calculating all genomic regions that were detected as a peak in at least 100 (15%) of our 669 samples. Regions neighboured within 100 bp were merged and very small regions of $<100$ bp were removed. In total, we obtained 26,384 H3K9Ac domains. A subset of 25,740 of these domains were located on autosomal chromosomes and used for subsequent analyses. To quantify H3K9Ac, we counted the number of reads falling into each H3K9Ac domain for each sample, divided by the width of each domain in kilobases and by the total number of mapped reads in each sample, and scaled this to obtain units of FPKMs. FPKM values were log-transformed before analysis. Data from 664 samples meeting the quality control criteria as above, and with full clinical data as described below, were included in these analyses.

**Assessment of clinical covariates.** We computed age at death from the self-reported date of birth and the date of death. We recorded sex at the time of the baseline interview.

The time of sunrise on the day of death was computed from the recorded date of death, and the latitude and longitude of the city in which each participant died.

Depressive symptoms were assessed with a 10-item version of the Centre for Epidemiologic Studies Depression scale[51].

Alzheimer's disease pathology was quantified as described previously[58,59]. Neurofibrillary tangles, diffuse plaques and neuritic plaques were visualized by Bielschowsky silver staining in sections from the frontal, temporal, parietal and entorhinal cortices and the hippocampus. As described in prior publications[59], a continuous global measure of the overall burden of Alzheimer's disease pathology was calculated by quantifying the highest density of each of neurofibrillary tangles, diffuse plaques and neuritic plaques per 1 mm$^2$ in sections from the frontal, temporal, parietal, hippocampal and entorhinal cortices of each participant, scaling these values using the s.d. of all participants, and then averaging across the four brain regions and three pathologies (tangles, diffuse and neuritic plaques) to generate a summary score. As described previously[58], for a categorical pathological diagnosis of Alzheimer's disease, cases were classified as no Alzheimer's disease, low-likelihood Alzheimer's disease, intermediate likelihood Alzheimer's disease or high likelihood Alzheimer's disease based on the National Institutes of Ageing-Reagan criteria[60]; a participant was considered to have a pathological diagnosis of Alzheimer's disease if their National Institutes of Ageing -Reagan classification was 'intermediate likelihood' or 'high likelihood'.

**Statistical analyses of diurnal and seasonal rhythmicity.** We concurrently characterized independent diurnal and seasonal patterns in the expression of each of the 18,709 genes using cosine functions, considering each transcript as a function date of death and time of death relative to sunrise on the date of death ('ZT', which is reflective of the timing of natural light exposure), adjusted for age at death, sex, post-mortem interval, the burden of Alzheimer's disease pathology, the presence/absence of depressive symptoms and technical covariates (batch, RNA integrity score for RNA-seq data, sample cross-correlation for H3K9Ac ChIP-seq data) as follows:

$$E(Y) = \beta_0 + \sum_{i=1}^{n} \beta_i x_i + A_d \cos(t - \phi_d) + A_s \cos(d - \phi_s) \qquad (1)$$

where $t$ is the ZT, and $d$ is the date of death, $A_d$ is the amplitude of diurnal rhythmicity, $A_s$ is the amplitude of seasonal rhythmicity, $\phi_d$ is the time of the acrophase of diurnal rhythmicity and $\phi_s$ is the date of the acrophase of seasonal rhythmicity. For these analyses, the diurnal period was fixed at 24 h and the seasonal period was fixed at 365 days. This is a limitation of any study design where each individual contributes only one data point to each 24-h or 365-day sampling period. All times of death were converted to radians ($2\pi$ radians $= 24$ h; 0 radians $=$ sunrise) for analysis and then converted back to hours for the purposes of visual representation. Dates of death were similarly converted to radians ($2\pi$ radians $= 365$ days; 0 radians $=$ January 1). In secondary analyses, we repeated all analyses using two alternative reference times: local clock time (with ZT0 $=$ midnight), to which the timing of artificial light exposure is linked, and the midpoint between sunset and sunrise, which is invariant across the seasons. For computational efficiency, we fit equivalent linearized models of the form

$$E(Y) = \beta_0 + \sum_{i=1}^{n} \beta_i x_i + \beta_{d_{cos}} \cos(t) + \beta_{d_{sin}} \sin(t) + \beta_{s_{cos}} \cos(d) + \beta_{s_{sin}} \sin(d)$$
$$(2)$$

and, $A_d$, $\phi_d$, $A_s$ and $\phi_s$ from equation (1) were calculated using the formulae

$$A_d = \sqrt{\beta_{d_{cos}}^2 + \beta_{d_{sin}}^2} \qquad (3)$$

$$A_s = \sqrt{\beta_{s_{cos}}^2 + \beta_{s_{sin}}^2} \qquad (4)$$

$$\phi_d = \text{atan2}\left(\frac{\beta_{d_{sin}}}{\beta_{d_{cos}}}\right) \qquad (5)$$

$$\phi_s = \text{atan2}\left(\frac{\beta_{s_{sin}}}{\beta_{s_{cos}}}\right) \qquad (6)$$

To quantify the contribution of diurnal rhythmicity to the model fit in equation (2) for each transcript, independent of the effects of seasonal rhythmicity, we compared the residual sum of squares for equation (2) to that of a reduced model without the diurnal terms

$$E(Y) = \beta_0 + \sum_{i=1}^{n} \beta_i x_i + \beta_{s_{cos}} \cos(d) + \beta_{s_{sin}} \sin(d) \qquad (7)$$

and determined the $F$-statistic

$$F_d = \frac{\left(\text{RSS}_{Eq7} - \text{RSS}_{Eq2}\right)/2}{n - 17} \qquad (8)$$

The greater the contribution of diurnal rhythmicity to the overall model fit, the greater the value of $F_d$. In similar way, we quantified the contribution of seasonal rhythmicity to the model fit in equation (2) by comparing the residual sum squares for equation (2) to that of a reduced model without the seasonal terms and determined the corresponding $F$-statistic.

We then repeated the above analyses for each of the 42,873 individual isoforms, 25,740 autosomal H3K9Ac peaks and 420,132 autosomal DNA methylation sites.

We generated a list of putatively diurnal genes by calculating the corresponding $P$ value and setting a threshold of $P < 0.05$. We compared this list of putatively diurnally rhythmic genes to three other published data sets (the dorsolateral prefrontal cortex data from Li et al.[18], and the BA47 and BA11 data from Chen et al.[19]), examining for enrichment using the $\chi^2$-test. For the data from Li et al.[18], only data from the top 50 most rhythmic transcripts were available for comparison, and we considered only transcripts identified as diurnally rhythmic in the dorsolateral prefrontal cortex at $P < 0.05$. For the data from Chen et al.[19], we considered all transcripts considered diurnally rhythmic at $P < 0.05$ in BA11 or BA47. Next, we compared the time of peak expression of core circadian clock genes (as annotated by GENECARDS[61]) between our data and these two other data sets by calculating circular correlation coefficients.

Next, we quantified the degree of diurnal rhythmicity across all 18,709 genes or 42,873 transcripts by computing the median $F_d$ across all transcripts in the observed data ($F_{d\_observed}$). To compute an empiric $P$ value, we compared this to the median $F_d$ across all transcripts in each of 10,000 null data sets ($F_{d\_null}$) generated by randomly shuffling the times of death, and determined the proportion of null data sets for which $F_{d\_null} > F_{d\_observed}$. We then did the same for seasonal rhythmicity, using null data sets generated by randomly shuffling dates rather than times of death. Then, we repeated these analyses for the 25,740 autosomal H3K9Ac peaks and 420,132 autosomal DNA methylation sites.

We next examined for patterns in the timing of diurnal and seasonal rhythms of RNA expression. By visual inspection, and in keeping with prior work[19,26], transcript diurnal and seasonal acrophase times were bimodally distributed. Based on this, we used self-organizing maps with toroidal grids to empirically define two diurnal and two seasonal clusters, and to classify each transcript into these clusters. The resulting diurnal clusters were roughly centred in the morning ($\sim$ZT0) and in the evening ($\sim$ZT12), and the resulting seasonal clusters were roughly centred in the spring and the fall. We examined the association between diurnal and seasonal classification by calculating the $\chi^2$-statistic for the corresponding $2 \times 2$ contingency table using the observed data ($\chi^2_{obs}$), comparing this to the corresponding $\chi^2$-statistics computed by repeating these procedures on 10,000 empiric null data sets generated by shuffling both the times and dates of death ($\chi^2_{null}$), and calculating an empiric $P$ value by determining the proportion of 10,000 null data sets in which $\chi^2_{null} > \chi^2_{obs}$. We then repeated this analysis for the H3K9Ac and DNA methylation data, except that for the DNA methylation data, we considered the time of the nadir rather than acrophase of methylation, as hypomethylation rather than hypermethylation is classically associated with transcription. Finally, as a sensitivity analysis, we repeated these analyses considering only those transcripts/peaks/sites that were both seasonally rhythmic and diurnally rhythmic at $P < 0.05$.

We previously showed that for rhythms of DNA methylation, the time of the diurnal acrophase varies depending on distance from the nearest TSS[27]. We investigated whether this was also true for diurnal rhythms of H3K9Ac by dividing H3K9Ac peaks into two groups—those within 2,000 bp of active TSS, and those more than 2,000 bp away from such TSS, where we defined active as corresponding to a transcript expressed in more than 90% of our samples. We compared the angular distributions of the diurnal acrophases of the H3K9Ac peaks in these two groups in the observed data by computing the Mardia–Watson–Wheeler

$W$-statistic ($W_{d\_observed}$) and compared this to the corresponding $W$-statistic for each of 10,000 null data sets generated by randomly shuffling times of death ($W_{d\_null}$), and calculated an empiric $P$ value by determining the proportion of 10,000 null data sets where $W_{d\_null} > W_{d\_observed}$. We then did the same for the angular distribution of seasonal acrophases, using null data sets generated by shuffling dates of death. Next, we examined whether the classification of individual peaks into diurnal/seasonal clusters differed between the two groups of H3K9Ac peaks by calculating the $\chi^2$-statistic, $\chi^2_{observed}$, for the corresponding $4 \times 2$ contingency table (4 possible seasonal/diurnal classes $\times$ 2 groups of H3K9Ac peaks), comparing this to the corresponding $\chi^2$-statistics generated by repeating the above procedure on 10,000 null data sets generated by shuffling times and dates of death, and calculating an empiric $P$ value by determining the proportion of 10,000 null data sets in which $\chi^2_{null} > \chi^2_{observed}$. We then repeated the above procedure for DNA methylation sites. Finally, as a sensitivity analysis, we repeated these analyses considering only those transcripts/peaks/sites that were both seasonally and diurnally rhythmic at $P < 0.05$.

Both H3K9Ac and DNA methylation are thought to influence transcription. Therefore, we examined associations between the timing of rhythms of transcript abundance and the timing of rhythms of H3K9Ac and DNA methylation at nearby sites. We first examined diurnal rhythms of H3K9Ac, using the 471 samples where both sets of data were available.

We considered two groups of H3K9Ac peaks: those within 2 kb of the TSS of transcripts empirically classified as morning acrophase, and those within 2 kb of the TSS of transcripts empirically classified as evening acrophase, and compared the angular distributions of their diurnal acrophases using the Mardia–Watson– Wheeler $W$-statistic as above, computing an empiric $P$ value by comparing this to the equivalent $W$-statistic calculated on 10,000 null data sets generated by randomly shuffling times of death. We then determined whether the two groups differed in the proportion of H3K9Ac peaks classified as morning or evening by calculating the corresponding $\chi^2$ statistic, and computing an empiric $P$ value by comparison with the equivalent $\chi^2$-statistic calculated from 10,000 null data sets generated by shuffling times of death. After considering diurnal rhythms of H3K9Ac as above, we then repeated this analysis considering seasonal rhythms. Next, we repeated these analyses for DNA methylation sites using the 527 samples with both RNA-seq and DNA methylation data. Finally, as a sensitivity analysis, we repeated these analyses considering only those transcripts/peaks/sites that were seasonally or diurnally rhythmic at $P < 0.05$.

The local transcription factor environment has an important influence on the circadian timing of transcription in model systems[31]. Therefore, we examined the association between local transcription factor-binding sites on the timing of diurnal and seasonal acrophases/nadirs of transcript expression, H3K9Ac and DNA methylation. To do so, we used genome-wide-annotated binding sites for 161 transcription factors from the ENCODE project[32–34]. We considered a TSS, H3K9Ac peak or DNA methylation site to be locally associated with a transcription factor if it overlapped with one of its ENCODE-annotated binding sites, or was within 2,000 bp of it. We empirically classified transcripts, H3K9Ac peaks and DNA methylation sites into diurnal and seasonal clusters as above.

We then used logistic regression models of the form

$$\text{logit(spring versus fall)} = \beta_1 TF_1 + \beta_2 TF_2 + \cdots + \beta_{161} TF_{161} \quad (9)$$

to examine the independent association of the local presence of binding sites for each of the 161 ENCODE transcription factors with the odds of having a spring versus fall or evening versus morning transcript, H3K9Ac peak or DNA methylation site. We estimated uncorrected $P$ values and analysis-wide FDRs by comparing the effect estimates above to those generated from 10,000 null data sets generated by randomly shuffling times or dates of death. We identified a set of candidate transcription factors potentially involved in regulating diurnal and/or seasonal rhythmicity if their binding sites had an FDR $< 0.05$ for one of diurnal or seasonal rhythmicity for any of RNA, H3K9Ac or DNA methylation. As a sensitivity analysis, we repeated this procedure considering only transcripts, H3K9Ac peaks or DNA methylation sites with $P < 0.05$ for diurnal or seasonal rhythmicity, and calculated Spearman's correlation coefficients relating the transcription factor effect estimates estimated on the basis of all transcripts/ H3K9Ac peaks/DNA methylation sites, and those estimated on the basis of only those transcripts/H3K9Ac peaks/DNA methylation sites with $P < 0.05$ for diurnal or seasonal rhythmicity.

Next, to examine for shared regulatory effects of these transcription factor-binding sites on diurnal and seasonal rhythmicity, we plotted their estimated coefficients for diurnal versus seasonal rhythmicity for each of transcript expression, H3K9Ac and DNA methylation, and calculated Spearman's correlation coefficients. Next, to examine for shared regulatory effects of these transcription factor-binding sites on transcript and epigenetic diurnal rhythms, we calculated Spearman's correlation coefficients relating to their effects on diurnal transcript versus epigenetic rhythms (H3K9Ac and DNA methylation). We then repeated this for seasonal rhythms.

Brain disorders such as Alzheimer's disease[35] have been shown to have an impact on physiological markers of circadian rhythmicity. To examine the impact of Alzheimer's disease on the amplitude and timing of rhythms of transcript expression, H3K9Ac and DNA methylation, we augmented equation (4) as follows:

$$\begin{aligned} E(Y) = \beta_0 + \sum_{i=1}^{n} \beta_i x_i + \left( \beta_{d_{cos}}^0 + \sum_{j=1}^{m} \beta_{d_{cos}}^j y_j + \beta_{d_{cos}}^{AD} AD \right) \cos(t) \\ + \left( \beta_{d_{sin}}^0 + \sum_{j=1}^{m} \beta_{d_{sin}}^j y_j + \beta_{d_{sin}}^{AD} AD \right) \sin(t) \\ + \left( \beta_{s_{cos}}^0 + \sum_{j=1}^{m} \beta_{s_{cos}}^j y_j + \beta_{s_{cos}}^{AD} AD \right) \cos(d) + \left( \beta_{s_{sin}}^0 + \sum_{j=1}^{m} \beta_{s_{sin}}^j y_j + \beta_{s_{sin}}^{AD} AD \right) \sin(d) \end{aligned}$$

$$(10)$$

where $y_1$ and $y_2$ are age at death and male sex.

Here, the relative amplitudes of diurnal and seasonal rhythmicity in individuals with Alzheimer's disease are given by

$$A_d^{AD} = \frac{\sqrt{\left( \beta_{d_{cos}}^0 + \beta_{d_{cos}}^{AD} \right)^2 + \left( \beta_{d_{sin}}^0 + \beta_{d_{sin}}^{AD} \right)^2}}{\overline{Y}} \quad (11)$$

$$A_s^{AD} = \frac{\sqrt{\left( \beta_{s_{cos}}^0 + \beta_{s_{cos}}^{AD} \right)^2 + \left( \beta_{s_{sin}}^0 + \beta_{s_{sin}}^{AD} \right)^2}}{\overline{Y}} \quad (12)$$

and the relative amplitudes of diurnal and seasonal rhythmicity in individuals without Alzheimer's disease are given by

$$A_d^{noAD} = \frac{\sqrt{\left( \beta_{d_{cos}}^0 \right)^2 + \left( \beta_{d_{sin}}^0 \right)^2}}{\overline{Y}} \quad (13)$$

$$A_s^{noAD} = \frac{\sqrt{\left( \beta_{s_{cos}}^0 \right)^2 + \left( \beta_{s_{sin}}^0 \right)^2}}{\overline{Y}} \quad (14)$$

Moreover, the acrophases of diurnal and seasonal rhythmicity in individuals with Alzheimer's disease are given by

$$\phi_d^{AD} = \text{atan2} \left( \frac{\beta_{d_{sin}}^0 + \beta_{d_{sin}}^{AD}}{\beta_{d_{cos}}^0 + \beta_{d_{cos}}^{AD}} \right) \quad (15)$$

$$\phi_s^{AD} = \text{atan2} \left( \frac{\beta_{s_{sin}}^0 + \beta_{s_{sin}}^{AD}}{\beta_{s_{cos}}^0 + \beta_{s_{cos}}^{AD}} \right) \quad (16)$$

and the acrophases of diurnal and seasonal rhythmicity in individuals without Alzheimer's disease are given by

$$\phi_d^{noAD} = \text{atan2} \frac{\beta_{d_{sin}}^0}{\beta_{d_{cos}}^0} \quad (17)$$

$$\phi_s^{noAD} = \text{atan2} \frac{\beta_{s_{sin}}^0}{\beta_{s_{cos}}^0} \quad (18)$$

To depict graphically the association between a pathological diagnosis of Alzheimer's disease and the timing of transcript rhythms, we classified each transcript into diurnal and seasonal classes as above, based on the model predicted acrophase for samples without Alzheimer's disease, and separately plotted the model-predicted diurnal and seasonal acrophase times for those with Alzheimer's disease. To more formally quantify the association between a pathological diagnosis of Alzheimer's disease and the timing of transcript rhythms, we calculated the median difference in predicted acrophase timing between individuals with and without Alzheimer's disease, first considering all transcripts and then considering separately transcripts in each of the four temporal classes. This procedure was repeated in 10,000 null data sets generated by shuffling times of death to generate empiric $P$ values. We then repeated this procedure for seasonal acrophases. To quantify the effect of Alzheimer's disease on the amplitude of transcript rhythms, we calculated the median difference in relative amplitude between individuals with and without Alzheimer's disease, first considering all transcripts, and then considering separately transcripts in each of the four temporal clusters. This procedure was repeated in 10,000 null data sets generated by shuffling times of death to generate empiric $P$ values. We then repeated this procedure for the relative amplitude of seasonal rhythms. As a sensitivity analysis, we then repeated the above considering only those transcripts with individual $P < 0.05$ for both diurnal and seasonal rhythmicity. Finally, we repeated the above analyses for the H3K9Ac peaks, and DNA methylation sites.

**Data availability.** RNA-seq data that support the findings of this study have been deposited in the synapse.org AMP-AD knowledge portal with the accession code syn3388564 (https://www.synapse.org/#!Synapse:syn3388564). H3K9Ac ChIP-seq data that support the findings of this study have been deposited in the synapse.org AMP-AD knowledge portal with the accession code syn4896408 (https://www.synapse.org/#!Synapse:syn4896408). DNA methylation data that support the findings of this study have been deposited in the synapse.org AMP-AD

knowledge portal with the accession code syn3157275 (https://www.synapse.org/#!Synapse:syn3157275).

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

## Acknowledgements

This research was supported in part by National Institutes of Health (http://www.nih.gov) grants P30AG10161 (to DAB), R01AG15819 (to DAB), R01AG17917 (to DAB) R01AG36042 (to DAB), R01AG36836 (to PLDJ), U01AG046152 (to PLDJ) and Canadian Institutes of Health Research (http: //www.cihr-irsc.gc.ca) grants MOP125934 (to ASPL), MMC112692 (to ASPL) and MSH136642 (to ASPL). Computations were performed on the General Purpose Cluster supercomputer at the SciNet HPC Consortium. SciNet is funded by the Canada Foundation for Innovation (http://www.innovation.ca) under the auspices of Compute Canada; the Government of Ontario (http://www.ontario.ca); Ontario Research Fund—Research Excellence (http://www.ontario.ca/business-and-economy/ontario-research-fund-research-excellence); and the University of Toronto (http://www.utoronto.ca). The funders had no role in study design, data collection and analysis, decision to publish or preparation of the manuscript.

## Author contributions

Conceptualization: A.S.P.L., P.L.D.J.; methodology: A.S.P.L., H.-U.K., J.X., P.L.D.J.; software: A.S.P.L., H.-U.K., S.A., J.X.; formal analysis: A.S.P.L., H.-U.K., S.A., L.B.C., L.Y.; investigation: H.-U.K., J.X., P.L.D.J.; resources: A.S.P.L., D.A.B., P.L.D.J.; writing—original draft: A.S.P.L., H.-U.K., J.X.; writing—review and editing: A.S.P.L., H.-U.K., L.Y., L.B.C., S.A., J.X., D.A.B., P.L.D.J.; supervision: A.S.P.L., D.A.B., P.L.D.J.; funding acquisition: P.L.D.J., D.A.B., A.S.P.L.

## Additional information

**Competing interests:** The authors declare no competing financial interests.

