## [Peer Review File · Nature Communications]

Reviewers' comments:

Reviewer #1 (Remarks to the Author):

This study by Lim et al sets out to examine diurnal and seasonal rhythms in gene expression and epigenetic marks in dorsolateral prefrontal cortex from 763 older individuals, including individuals with Alzheimer's disease. Fig. 1a shows expression data for *Arntl1* (*Bmal1*) from brain samples relative to time of death, with ZT0 being sunrise. First, their expression data show what appears to be a weak diurnal rhythm, with two nadirs at ZT2 and ZT18. There should be only one nadir, and given the large # of brains examined, the rhythm should be strong. Huda Akil's group published a seminal paper where a similar analysis was done with postmortem human brain samples (Li et al; ref 18 in Lim's study). Fig 1C in Li et al's PNAS study shows a striking rhythm in *Arntl* and several other circadian clock genes. Li et al used the same definition for ZT0 (= sunrise relative to time of death), and shows a single nadir at ~ZT6 for the same gene (*Arntl*). These discrepancies raise the question as to Lim et al's data quality and/or analytic approach. Lim et al do not show data for other genes that would be expected to cycle in human brain, further raising concern that data quality is low.

Lim et al could further validate their data by identifying cycling genes and compare to the list of cycling genes identified in Akil's study.

Regarding Alzheimer's data, Lim et al did not validate the quality of their expression data. At a minimum, it would be ideal to compare relative to several published postmortem Alzheimers disease expression data sets.

Reviewer #2 (Remarks to the Author):

This is an interesting manuscript using sophisticated analyses to investigate diurnal and seasonal rhythmicity in the transcriptome and epigenome in human neocortex in brains of Alzheimer and non-Alzheimer patients. In my view the analysis of diurnal rhythmicity may be fundamentally flawed because the analyses were conducted relative to ZT with ZT0 defined as sunrise. In industrialised societies the sleep-wake cycle (and the phase of the circadian clock) is determined by clock time, not by sunrise! (We don't wake up several hours earlier in summer than in winter. In addition, our light exposure is to a large extent determined by artificial light which is of course linked to the sleep-wake cycle. Furthermore, the duration of the photoperiod changes with season and this implies that ZT12 is only the onset of darkness during the equinoxes. By analysing the diurnal rhythmicity to ZT0 and by linking ZT0 to sunrise a confound between diurnal rhythmicity and seasonal rhythmicity is introduced and this may explain some of the dependencies and cluster identified. Leading circadian biologists have recognised this problem (see Daan et al 2002 PMID: 12002157) and proposed a solution: Analyse data relative to the midpoint of the dark period which doesn't change much with season, except for the influence of Daylight savings time, if one takes the natural light-dark cycle as the reference point. [One could also take the midpoint of the sleep period as reference point]. So my request is that the data be reanalysed as suggested. I will then be more than happy to evaluate this interesting manuscript again.

Reviewer #3 (Remarks to the Author):

In this study Lim and colleagues profile post-mortem human brains with date of death varying throughout the day and year to identify changes in gene expression and epigenetic marks that correlate with diurnal and seasonal variation. The authors also demonstrate that these patterns of

correlation are disrupted in brain tissue from patients with Alzheimer's disease. This is likely the first study to examine seasonal changes in gene expression and epigenetic modifications in the human brain in a genome-wide manner. Therefore it should provide an important contribution to our understanding of human brain gene expression as well as how these patterns of gene expression can vary across time and become disrupted in disorders such as Alzheimer's disease.

I have two main concerns.

The first concern is with the transcription factor analysis. The ENCODE dataset is based upon ChIP data from transformed human cell lines. I am not sure comparisons to human brain tissue makes sense. A better comparison might be to use the ROADMAP consortium brain tissue dataset, however, that dataset only has epigenetic marks not transcription factor data. In general, I am not sure the transcription factor analysis adds much to the overall study so I might suggest not including these data.

The second concern is with regards to the results relevant to Alzheimer's disease. The authors clearly find shifts in AD brain when examining gene expression. However, there are only weak changes in the methylation and acetylation data. In the control dataset, the authors make big claims about linking expression to acetylation and methylation in a diurnal and seasonal manner. They also mention the "relationship between the timing of diurnal rhythms of histone modification and diurnal rhythms of transcription." How then can the authors dissociate these correlations in the AD brain when only the transcriptome data is significantly shifted? The authors only mention "other compensatory factors" — can they elaborate? It's also a bit concerning that the authors' previous work (Ref #27) is the impetus for this study but now the authors find that some of the previously reported effects go away when making the proper covariate adjustments.

Minor concerns:

The authors state: "We found many genes showing diurnal (Fig. 1a) and seasonal (Fig. 1b) rhythmicity." But only show 1 gene (ARNTL) in the figure. Could the authors clarify whether they are showing the aggregate of many genes or just ARNTL and adjust the text accordingly.

The authors state: "For the DNA methylome, we examined the timing of the nadir rather than acrophase of methylation as hypo rather than hyper methylation is classically associated with transcription." Given that the authors took the time to explain their reasoning here, I would assume that they have already looked at the acrophase rather than nadir for the DNA methylome. It would be nice to see that result even if it's negative (potential to repress repressors, etc).

It would be good to show that there is no relationship between time of death and season within the samples themselves. Mostly because almost all of the results show an association between TOD and season and the authors attribute it to gene expression / epigenetic changes. It is unlikely that there is any relation between average time of day and when it is during the year, but may be worth checking. The idea would be to rule out anything unusual (influx of new doctors in the fall -> differences in the way TOD is recorded or something).

Did the authors correct for daylight savings? Most people don't adjust their schedule dramatically to accommodate for the change in light cycle, but I would be curious if this affects any of the data especially subtle shifts.

The authors state that the data do not support photoperiod being a "primary driver" of seasonal gene expression. Are the data points sufficiently "covered" across all time periods to have enough

confidence regarding peak and trough calling?

The authors state: "The time of sunrise on the day of death was computed from the recorded date of death" How much of the results could be driven by the length of time between sunrise and a given time during the day, which is then driven by the season on that date. There is of course a lot to be said about light being one of the strongest drivers of circadian expression of genes, but humans are more complicated than a model system organism with little external and internal behavioral pressures.

Figures 1A and B are double plotted, while the modeled panels 1C and D are single plotted. It might allow for easier comparison if all panels are either single or double plotted. Also I'm not sure why the authors chose to plot a full month of daily rhythms in D instead of a more typical 48h plot as in panel A.

RESPONSE TO REVIEWER QUESTIONS

REVIEWER 1

R1.1 This study by Lim et al sets out to examine diurnal and seasonal rhythms in gene expression and epigenetic marks in dorsolateral prefrontal cortex from 763 older individuals, including individuals with Alzheimer's disease. Fig. 1a shows expression data for Arntl1 (Bmal1) from brain samples relative to time of death, with ZT0 being sunrise. First, their expression data show what appears to be a weak diurnal rhythm, with two nadirs at ZT2 and ZT18. There should be only one nadir, and given the large # of brains examined, the rhythm should be strong. Huda Akil's group published a seminal paper where a similar analysis was done with postmortem human brain samples (Li et al; ref 18 in Lim's study). Fig 1C in Li et al's PNAS study shows a striking rhythm in Arntl and several other circadian clock genes. Li et al used the same definition for ZT0 (= sunrise relative to time of death), and shows a single nadir at ~ZT6 for the same gene (Arntl). These discrepancies raise the question as to Lim et al's data quality and/or analytic approach. Lim et al do not show data for other genes that would be expected to cycle in human brain, further raising concern that data quality is low.

We thank the reviewer for the opportunity to highlight published and new data that address the question of diurnal rhythmicity of clock gene expression in our samples. In a prior publication (Lim et al, 2013) we determined clock gene expression in a subset of the samples in this study using a microarray platform, demonstrated robust rhythmicity of several clock genes, and showed good concordance with data from model organisms. In addition, we now supplement these published microarray data, and our original isoform-wise analysis of the RNA-seq data in this study, with a new gene-wise analysis of our RNA-seq data (Supplemental Table 1 of the revised manuscript). These data show robust diurnal rhythms of clock gene expression, similar to our previous report, irrespective of whether the reference time is sunrise (Fig. 2 of the revised manuscript, attached below), clock time (Supplemental Fig. 2 of the revised manuscript) or the midpoint of the dark period (Supplemental Fig. 3 of the revised manuscript). This highlights that a) rhythms of clock gene expression in our samples are independent of the platform used to assess gene expression (microarray vs. RNA-seq) and that b) they are also robust to the definition of ZT=0. Moreover, the p-values for diurnal rhythmicity of known clock genes are similar to the data of Li et al cited by Reviewer 1 (Li et al, 2013), and the timing of their peak expression is highly correlated with the data of Li et al, and with the data of another large published study of human neocortical gene expression (Chen et al, 2016) with $R > 0.95$, slopes near 1, and a vertical phase shift in keeping with the age difference between our cohort and these other cohorts. Taken together, these data highlight the robustness of our dataset for measuring rhythms of clock gene expression, and its concordance with other large human brain diurnal rhythm datasets.

Text pertaining to these analyses and results can be found on page 5 line 99, page 5 lines 116-117, and; page 14 lines 318-324 of the revised Discussion section; page 20 lines 493-494 of the revised Methods section; and page 46 paragraph 4 of the revised Supplementary Methods section.

REFERENCES

- Chen CY, Logan RW, Ma T, Lewis DA, Tseng GC, Sibille E, et al. Effects of aging on circadian patterns of gene expression in the human prefrontal cortex. *Proc Natl Acad Sci U S A*. 2015.
- Li JZ, Bunney BG, Meng F, Hagenauer MH, Walsh DM, Vawter MP, et al. Circadian patterns of gene expression in the human brain and disruption in major depressive disorder. *Proc Natl Acad Sci U S A*. 2013;110(24):9950-5.
- Lim AS, Myers AJ, Yu L, Buchman AS, Duffy JF, De Jager PL, et al. Sex difference in daily rhythms of clock gene expression in the aged human cerebral cortex. *J Biol Rhythms*. 2013;28(2):117-29.

Figure 2: Diurnal and seasonal rhythms of clock gene expression. (a-b): Relative expression by time of death (a) and month of death (b) for the several genes known to be involved in the regulation of the mammalian circadian clock. Data plotted in 4-hour (a) or 2-month (b) bins. ZT0 = sunrise. Dots indicate means and bars indicate standard errors of the mean. Data are double plotted. Red lines indicate best fit cosine curve. P-values for diurnal (a) or seasonal (b) rhythmicity are as calculated as described in the text using a model considering diurnal and seasonal rhythmicity concurrently, and adjusted for demographic and methodological covariates. (c): Correlation between the peak expression times of known circadian clock genes in our dataset (y-axis) and published human prefrontal cortex datasets (x-axis). Red line indicates best-fit linear regression.

R1.2 Lim et al could further validate their data by identifying cycling genes and compare to the list of cycling genes identified in Akil's study.

In a gene-wise analysis of our data (Supplemental Table 1 of the revised manuscript), we found that ~9% of genes in the dorsolateral prefrontal cortex show diurnal rhythmicity at $p < 0.05$, a similar proportion as found in the studies of Li et al, and Chen et al. Many canonical clock genes (see response to R1.1 above) were among the most rhythmic genes in all three datasets. Moreover, the set of diurnally rhythmic genes in our dorsolateral prefrontal cortex data was highly enriched for genes identified as diurnally rhythmic in adjacent Brodmann's area 47 ($\chi^2=11.8$, $p=5.9 \times 10^{-4}$) and Brodmann's area 11 ($\chi^2=18.0$, $p=2.2 \times 10^{-5}$) in the Chen et al dataset, or identified as among the top 50 diurnally rhythmic genes in the Li et al dorsolateral prefrontal cortex dataset ($\chi^2=21.4$, $p=3.7 \times 10^{-6}$).

Text pertaining to these analyses can be found on page 6 lines 130-139 of the revised Results section; page 14 lines 320-324 of the revised Discussion section; and page 22 lines 516-523 of the revised Methods section.

REFERENCES

Chen CY, Logan RW, Ma T, Lewis DA, Tseng GC, Sibille E, et al. Effects of aging on circadian patterns of gene expression in the human prefrontal cortex. Proc Natl Acad Sci U S A. 2015.
Li JZ, Bunney BG, Meng F, Hagenauer MH, Walsh DM, Vawter MP, et al. Circadian patterns of gene expression in the human brain and disruption in major depressive disorder. Proc Natl Acad Sci U S A. 2013;110(24):9950-5.

R1.3 Regarding Alzheimer's data, Lim et al did not validate the quality of their expression data. At a minimum, it would be ideal to compare relative to several published postmortem Alzheimers disease expression data sets.

We compared gene expression in our data to that of the largest previous AD brain microarray dataset (Zhang et al, 2013). We first used a consensus clustering algorithm (Gaiteri et al, 2015) to identify modules of co-expressed genes both in our data and in the Zhang dataset. We then assessed the strength of the association of each module with a pathological diagnosis of Alzheimer's disease in both datasets. As can be seen in the figure below, there is good concordance in both the strength and direction of module-pathologic AD associations between our data and the Zhang dataset, highlighting the consistency of our data with other large human brain AD datasets.

REFERENCES

Gaiteri C, Chen M, Szymanski B, Kuzmin K, Xie J, Lee C, et al. Identifying robust communities and multi-community nodes by combining top-down and bottom-up approaches to clustering. Scientific reports. 2015;5:16361.
Zhang B, Gaiteri C, Bodea LG, Wang Z, McElwee J, Podtelezchnikov AA, et al. Integrated systems approach identifies genetic nodes and networks in late-onset Alzheimer's disease. Cell. 2013;153(3):707-20.

REVIEWER 2

R2.1 This is an interesting manuscript using sophisticated analyses to investigate diurnal and seasonal rhythmicity in the transcriptome and epigenome in human neocortex in brains of Alzheimer and non-Alzheimer patients. In my view the analysis of diurnal rhythmicity may be fundamentally flawed because the analyses were conducted relative to ZT with ZT0 defined as sunrise. In industrialised societies the sleep-wake cycle (and the phase of the circadian clock) is determined by clock time, not by sunrise! (We don't wake up several hours earlier in summer than in winter. In addition, our light exposure is to a large extent determined by artificial light which is of course linked to the sleep-wake cycle. Furthermore, the duration of the photoperiod changes with season and this implies that ZT12 is only the onset of darkness during the equinoxes. By analysing the diurnal rhythmicity to ZT0 and by linking ZT0 to sunrise a confound between diurnal rhythmicity and seasonal rhythmicity is introduced and this may explain some of the dependencies and cluster identified. Leading circadian biologists have recognised this problem (see Daan et al 2002 PMID: 12002157) and proposed a solution: Analyse data relative to the midpoint of the dark period which doesn't change much with season, except for the influence of Daylight savings time, if one takes the natural light-dark cycle as the reference point. [One could also take the midpoint of the sleep period as reference point]. So my request is that the data be reanalysed as suggested. I will then be more than happy to evaluate this interesting manuscript again.

We thank the reviewer for the suggestion. We agree that because free-living humans in industrialized societies are exposed to both natural and artificial light, each with their own schedules and seasonal dependencies, the decision of which reference time to use as ZT0 is much more complicated than for model organisms living in laboratory environments, and that there is potential for confounding. We initially used sunrise as ZT0 because this is what has been generally done in the other human circadian gene expression studies (e.g. Li et al, 2013; Chen et al, 2016), is what is generally done in the animal literature (where ZT0 is the laboratory “lights on” time), and reflects the timing of natural light exposure, which is an important source of light for humans even in industrialized societies. However, as suggested by the reviewer, to provide reassurance that our results are not solely due to seasonal variation in the relative timing of sunrise or in the relative timing of artificial light exposure, we have re-analysed our entire dataset using two additional definitions of ZT0: 1) the mid-point of the dark period, the timing of which is largely invariant across seasons (Daan et al, 2002) and 2) local clock time, to which the timing of artificial (rather than natural) light exposure is linked, and which shifts with daylight savings time. The results of these secondary analyses are in Supplemental Figs. 5-6 of the revised manuscript (attached below) as well as Supplemental Figs. 2-3, 8-9, 11-12, 15-16, and 19-20, of the revised manuscript, all of which show that our results are essentially unchanged irrespective of the specific choice of ZT0, aside from the expected y-axis shifts reflecting the average time separating sunrise from midnight, or separating sunrise from the midpoint of the dark period.

In addition to the above, as correctly pointed out by the reviewer, the timing of sunset and sunrise varies with the seasons. Therefore, throughout the manuscript, rather than referring to “sunrise” and “sunset” acrophase/nadir transcripts, H3K9Ac peaks, and DNA methylation sites, we instead use the terms “morning” and “evening”

Text pertaining to these analyses can be found on page 5 lines 110-114, page 7 lines 151-154, page 9 lines 200-201, page 9 lines 207-208, page 10 lines 240-241, and page 12 lines 272-273 of the revised Results section; page 18 lines 424-430 of the revised Discussion section; page 21 lines 495-496 and 509-12 of the revised Methods section; and page 45 paragraph 3 of the revised Supplementary Methods section.

REFERENCES

- Chen CY, Logan RW, Ma T, Lewis DA, Tseng GC, Sibille E, et al. Effects of aging on circadian patterns of gene expression in the human prefrontal cortex. *Proc Natl Acad Sci U S A*. 2015.
- Daan S, Mrosovsky M, Roenneberg T. External time--internal time. *J Biol Rhythms*. 2002;17(2):107-9.
- Li JZ, Bunney BG, Meng F, Hagenauer MH, Walsh DM, Vawter MP, et al. Circadian patterns of gene expression in the human brain and disruption in major depressive disorder. *Proc Natl Acad Sci U S A*. 2013;110(24):9950-5.

Figure 3: Diurnal and seasonal rhythmicity in the transcriptome, H3K9 acetylome, and DNA methylome. Diurnal rhythms referenced to sunrise. (a): Observed (red) vs. expected (black) median F-statistic for diurnal rhythmicity considering all transcripts together. Null distribution estimated by consideration of 10,000 empiric null datasets generated by randomly shuffling times of death. (b): as in a but for seasonal rhythms. (c-d): as in (a-b) but for H3K9Ac peaks. (e-f): as in (a-b) but for DNA methylation sites. (g): Association between time of diurnal vs. seasonal acrophases. Each dot represents a single transcript. Coloured boxes depict empirically derived clusters. (h): Observed (red line) vs. expected (black bars) χ^2 statistic for association between timing of diurnal and seasonal rhythms. Expected distribution empirically derived from 10,000 permuted null datasets generated by randomly shuffling times and dates of death. (i-j): same as (g-h) but for H3K9Ac peaks. (k-l): same as for (g-h) but for the diurnal and seasonal nadirs of individual DNA methylation sites. See also Supplementary Fig. 4-6.

Supplementary Figure 5: Diurnal and seasonal rhythmicity in the transcriptome, H3K9 acetylome, and DNA methylome. Diurnal rhythms referenced to local clock time. Same as Fig. 3, but reanalysed with ZT0 = midnight, local clock time. (a): Observed (red) vs. expected (black) median F-statistic for diurnal rhythmicity considering all transcripts together. Null distribution estimated by consideration of 10,000 empiric null datasets generated by randomly shuffling dates of death. (b): as in a but for seasonal rhythms. (c-d): as in (a-b) but for H3K9Ac peaks. (e-f): as in (a-b) but for DNA methylation sites. (g): Association between time of diurnal vs. seasonal acrophases. Each dot represents a single transcript. Coloured boxes depict empirically derived clusters. (h): Observed (red line) vs. expected (black bars) χ^2 statistic for association between timing of diurnal and seasonal rhythms. Expected distribution empirically derived from 10,000 permuted null datasets generated by randomly shuffling times and dates of death. (i-j): same as (g-h) but for H3K9Ac peaks. (k-l): same as for (g-h) but for the diurnal and seasonal nadirs of individual DNA methylation sites.

Supplementary Figure 6: Diurnal and seasonal rhythmicity in the transcriptome, H3K9 acetylome, and DNA methylome. Diurnal rhythms referenced to mid point of dark period. Same as Fig. 3, but reanalysed with ZT0 = midpoint of dark period. (a): Observed (red) vs. expected (black) median F-statistic for diurnal rhythmicity considering all transcripts together. Null distribution estimated by consideration of 10,000 empiric null datasets generated by randomly shuffling dates of death. (b): as in a but for seasonal rhythms. (c-d): as in (a-b) but for H3K9Ac peaks. (e-f): as in (a-b) but for DNA methylation sites. (g): Association between time of diurnal vs. seasonal acrophases. Each dot represents a single transcript. Coloured boxes depict empirically derived clusters. (h): Observed (red line) vs. expected (black bars) χ^2 statistic for association between timing of diurnal and seasonal rhythms. Expected distribution empirically derived from 10,000 permuted null datasets generated by randomly shuffling times and dates of death. (i-j): same as (g-h) but for H3K9Ac peaks. (k-l): same as for (g-h) but for the diurnal and seasonal nadirs of individual DNA methylation sites.

REVIEWER 3

R 3.2 In this study Lim and colleagues profile post-mortem human brains with date of death varying throughout the day and year to identify changes in gene expression and epigenetic marks that correlate with diurnal and seasonal variation. The authors also demonstrate that these patterns of correlation are disrupted in brain tissue from patients with Alzheimer's disease. This is likely the first study to examine seasonal changes in gene expression and epigenetic modifications in the human brain in a genome-wide manner. Therefore it should provide an important contribution to our understanding of human brain gene expression as well as how these patterns of gene expression can vary across time and become disrupted in disorders such as Alzheimer's disease.

I have two main concerns.

The first concern is with the transcription factor analysis. The ENCODE dataset is based upon ChIP data from transformed human cell lines. I am not sure comparisons to human brain tissue makes sense. A better comparison might be to use the ROADMAP consortium brain tissue dataset, however, that dataset only has epigenetic marks not transcription factor data. In general, I am not sure the transcription factor analysis adds much to the overall study so I might suggest not including these data.

In model systems, the timing of circadian rhythms of gene expression, and accompanying rhythms of histone modification, is driven in large part by the local transcription factor environment. Moreover, many core circadian clock genes are themselves transcription factors, and covalent modification (e.g. phosphorylation), nuclear translocation, and proteolytic degradation of rhythm-associated transcription factor complexes are steps in rhythmic control that may potentially be therapeutically targeted. Thus, identification of transcription factors upstream of diurnal and seasonal epigenetic and transcriptional rhythms in the human brain is of both scientific and therapeutic interest.

We are not aware of any large-scale datasets using ChIP-seq on human brain tissue to generate a set of annotations for human neocortex similar to the ENCODE data. We agree with the reviewer that the ENCODE dataset, which was generated from human cell lines, is an imperfect source of annotations for transcription factor binding sites in post-mortem human neocortical tissue. Therefore, like all analyses seeking to extrapolate from cell lines to human tissues, our transcription factor binding site analysis may be best thought of as a hypothesis-generating exploratory analysis that will need further validation. Our results will allow targeted human neocortical transcription factor ChIP-seq experiments, and *in vitro* studies in human brain-derived cell lines to further elucidate transcription factors regulating diurnal and seasonal epigenetic and transcriptional rhythms in the human brain.

Text pertaining to the above can be found on page 11 lines 250-252 of the revised Results section and page 16 lines 375-381 of the revised Discussion section.

R3.3 The second concern is with regards to the results relevant to Alzheimer's disease. The authors clearly find shifts in AD brain when examining gene expression. However, there are only weak changes in the methylation and acetylation data. In the control dataset, the authors make big claims about linking expression to acetylation and methylation in a diurnal and seasonal manner. They also mention the "relationship between the timing of diurnal rhythms of histone modification an diurnal rhythms of transcription." How then can the authors dissociate these correlations in the AD brain when only the transcriptome data is significantly shifted? The authors only mention "other compensatory factors"—can they elaborate?

We thank the reviewer for giving us the opportunity to elaborate further on potential explanations for this finding. We do not think that an association between H3K9 acetylation, DNA methylation, and transcript abundance in normal situations precludes their dissociation in the context of Alzheimer's disease pathology. There are at least two possible explanations, broadly speaking, for this observation. First, rhythms of transcription reflect not only rhythms of DNA methylation and H3K9 acetylation, but also rhythms of a whole host of other epigenetic marks,

as well as transcriptional events such as RNA polymerase binding (Le Martelot et al, 2012; Koike et al, 2012; Menet et al, 2012). It is possible that by affecting these other processes, Alzheimer's disease pathology may alter the phase relationship between rhythms of H3K9 acetylation, DNA methylation, and RNA transcription. Second, a body of evidence suggests that post-transcriptional processes that influence RNA degradation may play an important role in determining the timing of rhythms of transcript abundance independent of rhythms of transcription per se, and in some cases may drive rhythms of transcript abundance even in the absence of rhythmic transcription (Le Martelot et al, 2012; Koike et al, 2012; Menet et al, 2012). MicroRNAs may play an important role in this process (Du et al, 2014) and rhythms of alternative splicing may also play a role (McGlincy et al, 2012). It is conceivable that Alzheimer's disease pathology may alter these post-transcriptional RNA processes, further de-coupling rhythms of H3K9 acetylation and DNA methylation from rhythms of transcript abundance.

Text pertaining to this is found on page 17 lines 406-415 of the revised Discussion section.

REFERENCES

- Du NH, Arpat AB, De Matos M, Gatfield D. MicroRNAs shape circadian hepatic gene expression on a transcriptome-wide scale. *eLife*. 2014;3:e02510.
- Koike N, Yoo SH, Huang HC, Kumar V, Lee C, Kim TK, et al. Transcriptional Architecture and Chromatin Landscape of the Core Circadian Clock in Mammals. *Science*. 2012.
- Le Martelot G, Canella D, Symul L, Migliavacca E, Gilardi F, Liechti R, et al. Genome-wide RNA polymerase II profiles and RNA accumulation reveal kinetics of transcription and associated epigenetic changes during diurnal cycles. *PLoS Biol*. 2012;10(11):e1001442.
- McGlincy NJ, Valomon A, Chesham JE, Maywood ES, Hastings MH, Ule J. Regulation of alternative splicing by the circadian clock and food related cues. *Genome Biol*. 2012;13(6):R54.
- Menet JS, Rodriguez J, Abruzzi KC, Rosbash M. Nascent-Seq reveals novel features of mouse circadian transcriptional regulation. *eLife*. 2012;1:e00011.

R3.4 It's also a bit concerning that the authors' previous work (Ref #27) is the impetus for this study but now the authors find that some of the previously reported effects go away when making the proper covariate adjustments.

Almost all of the results from our prior work (Lim et al, 2013) were replicated in this analysis. The only result that was statistically significant in the prior analysis but fell below the threshold of significance in this analysis is the effect of Alzheimer's disease pathology on the median timing of the nadir of DNA methylation. In our 2013 publication, we examined the impact of Alzheimer's disease pathology on the median timing of the nadir of the 21,842 highest-amplitude DNA methylation sites, referenced to midnight clock time, and unadjusted for season of death. We found that subjects with pathologic AD had a significant 1 hour 29 minute median delay in the methylation nadir. In the present study, we examined the impact of Alzheimer's disease pathology on the median timing of the nadir of all 420,132 DNA methylation sites, referenced to sunrise, and adjusted for season of death. We found an attenuated effect of Alzheimer's disease pathology on the median nadir of methylation (subjects with a pathological diagnosis of AD had a median delay of 27 minutes; $p=0.32$). At least some of this discrepancy likely reflects the different reference time frame (clock time vs. sunrise time). Indeed, in secondary analyses, when we used midnight clock time as the reference time (Supplemental Fig. 19i of the revised manuscript) the phase delay associated with Alzheimer's disease pathology approached statistical significance ($p=0.12$). However, given the strong association between seasonal and diurnal rhythms that we have shown in this study, it is also likely that some of the difference between the present study and our previous study reflects confounding of diurnal effects by seasonal effects, highlighting the importance of adjusting for seasonal effects when examining diurnal rhythms.

Text pertaining to the above can be found on page 13 lines 298-301; and page 167 lines 394-401 and page 17 lines 391-393 of the revised Discussion section.

REFERENCES

Lim AS, Srivastava GP, Yu L, Chibnik LB, Xu J, Buchman AS, et al. 24-hour rhythms of DNA methylation and their relation with rhythms of RNA expression in the human dorsolateral prefrontal cortex. *PLoS genetics*. 2014;10(11):e1004792.

Minor concerns:

R3.5 The authors state: “We found many genes showing diurnal (Fig. 1a) and seasonal (Fig. 1b) rhythmicity.” But only show 1 gene (ARNTL) in the figure. Could the authors clarify whether they are showing the aggregate of many genes or just ARNTL and adjust the text accordingly.

We apologize for the lack of clarity. We have modified Figure 1 from the original manuscript (which is now Figure 2 in the revised manuscript) to show multiple individual clock genes, each in its own panel, as attached above in our response to comment R1.1.

R3.6 The authors state: “For the DNA methylome, we examined the timing of the nadir rather than acrophase of methylation as hypo rather than hyper methylation is classically associated with transcription.” Given that the authors took the time to explain their reasoning here, I would assume that they have already looked at the acrophase rather than nadir for the DNA methylome. It would be nice to see that result even if it's negative (potential to repress repressors, etc).

In our analyses, we parameterized our data using cosine curves with timing of the acrophase exactly π radians from the timing of the nadir. Thus, from a statistical perspective, statistical tests performed on the time of the nadir vs. time of the acrophase will yield the same p-value. Similarly, with regard to the figures, figures plotting acrophase time rather than nadir time will be phase shifted by π radians. For the bulk of the analyses, this does not lead to interpretative changes. However, when Fig. 4i-j is so plotted (attached below) plotting acrophase time rather than nadir time does highlight that whereas evening RNAs seemed to be associated with evening nadir DNA methylation sites, fall RNAs seem to be associated with fall acrophase (rather than nadir) DNA methylation sites. In the interests of brevity, and because of the statistical equivalence of analysing by nadir time vs. acrophase time, we have not re-drawn all of our figures twice (once for nadir, and once for acrophase). However, new text discussing the question of nadir vs. acrophase can be found on page 8 line 175-176, and page 10 lines 236-238 of the revised Results section, and page 15 lines 357-358 of the revised Discussion section.

R3.7 It would be good to show that there is no relationship between time of death and season within the samples themselves. Mostly because almost all of the results show an association between TOD and season and the authors attribute it to gene expression / epigenetic changes. It is unlikely that there is any relation between average time of day and when it is during the year, but may be worth checking. The idea would be to rule out anything unusual (influx of new doctors in the fall -> differences in the way TOD is recorded or something).

We thank the reviewer for the suggestion. We have confirmed that there is indeed no meaningful association between the time and date of death (angular $R=-0.04$). This is depicted in Fig. 1 of the revised manuscript (attached below) and is discussed on page 5 lines 97-98 of the revised Results section.

R3.8 Did the authors correct for daylight savings? Most people don't adjust their schedule dramatically to accommodate for the change in light cycle, but I would be curious if this affects any of the data especially subtle shifts.

We thank the author for the comment. To address this question, as described in our response to R2.1 above, we have repeated all of our analyses using three different definitions of ZT0: local sunrise time, the midpoint of the dark period, and local clock time. The first two do not shift with daylight savings time, while the last one does. The vast majority of our results are essentially unchanged irrespective of which definition of ZT0 is used, suggesting that they are robust to daylight savings time effects

Text pertaining to these analyses can be found on page 5 lines 110-114, page 7 lines 151-154, page 9 lines 200-201, page 9 lines 207-208, page 10 lines 240-241, and page 12 lines 272-273 of the revised Results section; page 18 lines 424-430 of the revised Discussion section; page 21 lines 495-496 and 509-12 of the revised Methods section; and page 45 paragraph 3 of the revised Supplementary Methods section.

R3.9 The authors state that the data do not support photoperiod being a “primary driver” of seasonal gene expression. Are the data points sufficiently “covered” across all time periods to have enough confidence regarding peak and trough calling?

We have generated a new figure (Fig. 1 of the revised manuscript, attached above under our response to R3.7) showing the distribution of deaths around the 24-hour day, and across the 12 months. There is good coverage of all hours (never fewer than 15 deaths in any given hour) and all months (never fewer than 25 deaths in any given month).

R3.10 The authors state: “The time of sunrise on the day of death was computed from the recorded date of death” How much of the results could be driven by the length of time between sunrise and a given time during the day, which is then driven by the season on that date. There is of course a lot to be said about light being one of the strongest drivers of circadian expression of genes, but humans are more complicated than a model system organism will little external and internal behavioral pressures.

We agree that confounding by seasonal effects on sunrise time is a possibility. To exclude this possibility, as described in more detail in our response to R2.1 above, we have repeated all of our analyses using three alternative definitions of ZT0: local sunrise time, the midpoint of the dark period, and local clock time. In particular, the midpoint of the dark period is independent of seasonal effects. We show that our results are similar irrespective of reference time and are therefore independent of seasonal effects on sunrise time or the timing of artificial light exposure.

Text pertaining to these analyses can be found on page 5 lines 110-114, page 7 lines 151-154, page 9 lines 200-201, page 9 lines 207-208, page 10 lines 240-241, and page 12 lines 272-273 of the revised Results section; page 18 lines 424-430 of the revised Discussion section; page 21 lines 495-496 and 509-12 of the revised Methods section; and page 45 paragraph 3 of the revised Supplementary Methods section.

R3.11 Figures 1A and B are double plotted, while the modeled panels 1C and D are single plotted. It might allow for easier comparison if all panels are either single or double plotted. Also I'm not sure why the authors chose to plot a full month of daily rhythms in D instead of a more typical 48h plot as in panel A.

We apologize for the lack of clarity. We have revised Fig. 1 from the original manuscript, which is now Fig. 2 in the revised manuscript (attached under our response to R1.1 above), and have double plotted all time series panels.

REVIEWERS' COMMENTS:

Reviewer #1 (Remarks to the Author):

The authors adequately addressed my concerns.

Reviewer #2 (Remarks to the Author):

The authors have adequately addressed my concern with respect to the 'reference' time, i.e. sunrise, clock time, midpoint of dark period. My only remaining comment is that it may be helpful for future publications that the authors extend the discussion around the difficulties selecting a reference time in humans. They now mention artificial light but don't mention that in general sleep timing (and associated exposure to darkness) doesn't change season but is linked to clock time (and associated social schedules).

Reviewer #3 (Remarks to the Author):

The authors have addressed my concerns in this revision.

RESPONSE TO REVIEWER COMMENTS

2.1: The authors have adequately addressed my concern with respect to the 'reference' time, i.e. sunrise, clock time, midpoint of dark period. My only remaining comment is that it may be helpful for future publications that the authors extend the discussion around the difficulties selecting a reference time in humans. They now mention artificial light but don't mention that in general sleep timing (and associated exposure to darkness) doesn't change season but is linked to clock time (and associated social schedules).

We thank the reviewer for the suggestion. The revised discussion contains an expanded discussion of the difficulties selecting an appropriate reference time in humans (Discussion Page 18 Lines 436-445)

“Third, humans experience both natural and artificial light, the timing of which may change by season. Moreover sleep timing, which is a major influence on exposure to darkness, is linked to clock time and associated social schedules rather than natural light exposure. These considerations complicate the selection of an appropriate reference time for diurnal rhythms, and also the interpretation of diurnal and seasonal rhythmicity. However, our results were robust: results were consistent between our primary analytic approach using sunrise time (reflective of the timing of natural light exposure) as the reference time for diurnal analyses, and our secondary analyses using clock time (linked to artificial light, and to the timing of sleep and social schedules) or the midpoint of the dark period (invariant across seasons) as the reference time.”